



**Hygroscopic behavior of aerosols generated from solutions of 3-methyl-1,2,3-butanetricarboxylic**
**acid, its sodium salts, and its mixtures with NaCl**
Li Wu[1], Clara Becote[2,3,4], Sophie Sobanska[2], Pierre-Marie Flaud[3,4], Emilie Perraudin[3,4], Eric Villenave[3,4],
Young-Chul Song[1], Chul-Un Ro[1*]
[1]Department of Chemistry, Inha University, Incheon, South Korea
[2]Institut des Sciences Moléculaires, UMR CNRS 5255, University of Bordeaux, Talence, France
[3]University of Bordeaux, EPOC, UMR 5805, 33405 Talence cedex, France
[4]CNRS, EPOC, UMR 5805, 33405 Talence cedex, France
**Abstract**
Secondary organic aerosols (SOAs), which are formed and transformed through complex
physicochemical processes in the atmosphere, have attracted considerable attention over the past decades
because of their impacts on both climate change and human health. Recently, 3-methyl-1,2,3-
butanetricarboxylic acid (MBTCA), a low volatile, highly oxidized, secondary generation product of
monoterpenes, is one of the most relevant tracer compounds for biogenic SOAs. Therefore, MBTCA was
selected to understand its hygroscopic properties better. In addition, interactions between the organic acid
and inorganic components have been reported, which may alter their hygroscopic properties mutually. In
this study, laboratory-generated, micrometer-sized, pure MBTCA, mono-/di-/tri-sodium MBTCA salts,
and MBTCA-NaCl mixture aerosol particles of four mixing ratios (molar ratios = 1:1, 1:2, 1:3, and 2:1)
were examined systematically to observe their hygroscopic behavior by varying the relative humidity
(RH) from RH = ~95% to ~1% through a dehydration process, followed by a humidification process from
RH = ~1% to ~95%, using in-situ Raman microspectrometry (RMS) assembled with a see-through
impactor where the particles were deposited on a Si wafer. The hygroscopic behavior of pure MBTCA
and MBTCA-NaCl mixture aerosol particles of three mixing ratios (molar ratios = 1:1, 1:2, and 1:3) were
also examined using a levitation system mounted on in-situ RMS through a humidification process from

*Corresponding author. Tel.: +82 32 860 7676; Fax: +82 32 867 5604; E-mail: curo@inha.ac.kr





RH = ~10% to ~80% after a quenching process from droplets, followed by dehydration from RH = ~80%
to ~10%. The pure MBTCA droplets effloresced at RH = ~30-57.8% and did not dissolve until RH >
95%. The mono- and di-sodium MBTCA salt aerosols did not show clear efflorescence RH (ERH) and
deliquescence RH (DRH). In contrast, the tri-sodium MBTCA salt exhibited ERH = ~44.4-46.8% and
DRH = ~53.1%, during the hygroscopic experiment cycle. The mixture aerosols generated from solutions
of MBTCA:NaCl = 1:1 and 2:1 showed no visible ERH and DRH in the see-through impactor because of
the partial and total consumption of NaCl, respectively, through chemical reactions during the
dehydration process. The mixture particles with a 1:1 molar ratio in the levitation system exhibited a clear
DRH at ~71% and ERH at ~50%. This suggests less reaction between the mixtures and a larger portion
of NaCl remaining in the levitation system. The other mixtures of MBTCA:NaCl = 1:2 and 1:3 displayed
single-stage efflorescence and deliquescence at ERH = ~45-50% and DRH = ~74%, respectively, because
of the considerable amount of NaCl present in the mixture aerosols in both systems. Observations and
Raman analyses indicated that only monosodium MBTCA salt aerosols could be formed through a
reaction between MBTCA and NaCl. The reaction occurred more rapidly with a more elevated
concentration of either MBTCA or NaCl, and the controlling factor for the reactivity of the mixtures
depended mostly on the availability of $H^+$ dissociated from the MBTCA tricarboxylic acid. The lower
degree of reaction of the mixture particles in the levitation system might be caused by the relatively
airtight circumstance inside, i.e., the less release of HCl. In addition, the quenching process, i.e., the
starting point of the hygroscopicity experiments, induced the solidification of MBTCA, and further, a
slow reaction between MBTCA and NaCl. The study revealed that the interactions between the MBTCA
and NaCl could modify the properties of the organic acid in the atmosphere, leading to enhanced
capability of the probable heterogeneous chemistry in the aqueous aerosols.

**1. Introduction**

Chemical processes, such as gas-phase oxidations of airborne biogenic and anthropogenic volatile

organic compounds (VOCs) by ozone ($O_3$), hydroxyl radical (OH), and nitrate radical ($NO_3$), and their
condensed-phase reactions with preexisting aerosols, can promote the formation of increasingly oxidized
and less volatile secondary organic aerosols (SOAs). SOAs are a ubiquitous and dominant fraction of the





fine aerosol mass that exists as liquid, amorphous solid, semi-solid, and phase-separated aerosol particles
(Jang et al., 2002; Hallquist et al., 2009; Jimenez et al., 2009; Virtanen et al., 2010; Koop et al., 2011;
Bateman et al., 2015b; Shrivastava et al., 2015; Bernard et al., 2016; Pajunoja et al., 2016; Freedman,
2017; Shrivastava et al., 2017; Kim et al., 2018; Srivastava et al., 2018; Liu et al., 2019; Slade et al., 2019;
Song et al., 2019; Wu et al., 2019a). These aerosols are of critical importance because of their ability to
scatter and absorb solar radiation directly, to affect the number of CCN (cloud condensation nuclei)
through the formation of new particles and the growth of preexisting particles, and further impact the
climate and human health (Haywood and Boucher, 2000; Topping et al., 2013; Poschl and Shiraiwa, 2015;
Reid et al., 2018; Marsh et al., 2019). SOAs are highly dynamic, multiphase chemical systems with a
range of volatility and solubility and model simulations have claimed that the phase state of SOAs differs
according to the global locations and altitudes with an evolving relative humidity (RH), temperature, and
particle composition (Kroll and Seinfeld, 2008; Shiraiwa et al., 2017).
Oxidative products of biogenic VOCs, such as monoterpenes (e.g., α- and β-pinene), act as a dominant
source of SOAs as they have high emission rates on a global scale and give considerable SOA yields, and
they play a central role in new particle formation (Guenther et al., 1995; Lignell et al., 2013; Mutzel et
al., 2016; Holopainen et al., 2017). Carboxylic acid-containing organic compounds comprise a large
fraction of SOAs in the Northern Hemisphere (Yatavelli et al., 2015). An extremely low-volatile
tricarboxylic acid, 3-methyl-1,2,3-butanetricarboxylic acid (MBTCA, $C_8H_{12}O_6$), has become one of the
most relevant tracer compound for terpene SOAs (Jaoui et al., 2005; Szmigielski et al., 2007; Zhang et
al., 2010; Donahue et al., 2012; Müller et al., 2012; Lai et al., 2015; Sato et al., 2016). In addition, it is
also a few well-known compounds with a high O:C ratio that is formed in the oxidation of VOCs (Dunne
et al., 2016). MBTCA is a second or later generation reaction product from monoterpenes by the OH-
initiated oxidation of pinonic acid (PA) in the gaseous and aqueous phases and even at the air-water
interface (Müller et al., 2012; Praplan et al., 2012; Aljawhary et al., 2016; Enami and Sakamoto, 2016).
The MBTCA concentrations were found to be positively correlated with temperature because of the
enhanced photochemical production of PA by OH radicals with increasing temperature (Hu et al., 2008;
Zhang et al., 2010; Gómez-González et al., 2012; Miyazaki et al., 2012). A further reaction between
MBTCA and OH radicals can result in $CO_2$ loss (Kostenidou et al., 2018). MBTCA can also accelerate





the new particle formation by effectively stabilizing initial molecular clusters with or without sulfuric
acid (Donahue et al., 2013; Elm, 2019). MBTCA was first observed at the Amazon basin and in summer
aerosols from Ghent, Belgium (Kubátová et al., 2000; Kubátová et al., 2002). The compound was later
found in the USA (Jaoui et al., 2005), Europe (Fu et al., 2009; Kourtchev et al., 2009; Zhang et al., 2010;
Yasmeen et al., 2011; Gómez-González et al., 2012; Vogel et al., 2013; Kammer et al., 2018; Vlachou et
al., 2019), Japan (Miyazaki et al., 2012), the polar regions (Hu et al., 2013), China (Hu et al., 2008; Ding
et al., 2012; Li et al., 2013; Fu et al., 2014; Kang et al., 2018; Hong et al., 2019), and Australia (Cui et al.,
2019). In addition, it has been observed in forest, marine, mountainous, urban, and rural aerosols, with its
levels ranging from 0.03 to 100 ng/m$^3$, and the level was generally higher in the fine particle fraction than
in the coarse fraction (Zhang et al., 2010).
The ability of the aerosol particles to uptake water in the air is dependent on one of the most important
physicochemical properties, i.e., the hygroscopicity (Jimenez et al., 2009; Chu et al., 2014; Tang et al.,
2019). Hygroscopicity can help better understand the (i) aerodynamic properties, (ii) cloud-droplet
nucleation efficiency, (iii) optical properties, and (iv) physicochemical changes through complicated
heterogeneous chemical reactions of aerosol particles with various atmospheric gas-phase species.
MBTCA was predicted to partition significantly into aerosol-liquid-water (ALW) (Aljawhary et al.,
2016). Therefore, a study on the hygroscopic behavior of MBTCA is important for understanding its
phase states better when it interacts with water vapor at different RHs as well as its impacts on the
heterogeneous chemical reactions, atmospheric environment, and human health (Parsons et al., 2004;
Mikhailov et al., 2009; Bateman et al., 2015a; Freedman, 2017; Slade et al., 2019). Atmospheric particles
typically involve complex internal mixtures of organic and inorganic compounds (Shrivastava et al.,
2017; Karadima et al., 2019). The interactions between organic and inorganic compounds may alter the
chemical compositions of SOAs, which in turn affect their physicochemical properties, such as
hygroscopicity (Rudich et al., 2007; Wu et al., 2011; Wang et al., 2015; Jing et al., 2016; Wang et al.,
2018). Dicarboxylic acids (DCAs) can undergo reactions with inorganics, such as NaCl, resulting in Cl
depletion and HCl liberation (Ma et al., 2013; Li et al., 2017). On the other hand, the interactions between
tricarboxylic acids and inorganics have never been investigated.





In this study, in situ Raman microspectrometry (RMS) was used to examine the hygroscopic behavior,
evolution of the chemical composition, phase states, and microstructures, and chemical reactivity of
laboratory-generated, micrometer-sized aerosols generated from a pure MBTCA solution, mono-/di-/tri-
sodium MBTCA salt solutions, and MBTCA-NaCl mixture solutions. RMS was assembled with either a
see-through impactor, where the particles were deposited on a Si wafer, or a levitation system. The
particles on the Si wafer were exposed to a hygroscopic measurement cycle, where they experienced a
dehydration process first (by decreasing RH from ~95 to ~1%), followed by a humidification process (by
increasing RH from ~1 to ~95%). The particles in the levitation system experienced a humidification
process first (by increasing the RH from ~10 to ~80%) after quenching from droplets, followed by a
dehydration process (by decreasing RH from ~80 to ~10%). NaCl, one of the major components of marine
aerosols, was selected as the inorganic component as it was previously reported that organic acids
contributed significantly to Cl depletion through a reaction with NaCl (Laskin et al., 2012). In situ Raman
analysis could clearly identify MBTCA and its sodium salts during the hygroscopicity measurement
despite NaCl being Raman inactive. To the best of the authors' knowledge, this is the first study on the
hygroscopic behavior and chemical reactivity of MBTCA and its sodium salts thus far. The results are
expected to promote more precise thermodynamic models (Clegg et al., 2003). The phase transitions were
observed by monitoring the size changes together with the Raman spectra evolutions of the aerosol
particles as a function of the RH. RMS can provide the aerosol compositions, water contents, molecular
interactions, and particle-phase states sensitively. Such data can help understand the hygroscopic behavior
of complex aerosol particles better (Lee et al., 2008; Li et al., 2017; Wang et al., 2017). The molecular
characterization of organic aerosols can provide better insights into the potential mechanisms of SOA
formation and transformation (or aging) (Hallquist et al., 2009). Scanning electron microscopy
(SEM)/energy-dispersive X-ray spectroscopy (EDX) mapping was used to examine the elemental
composition distribution in effloresced particles.

**2. Experimental Section**
**2.1 Sample preparation**





Pure 0.3 M solutions of NaCl (>99.9% purity, Sigma-Aldrich) and MBTCA (98%, Toronto Research
Chemicals, TCR) were prepared. The mixture solutions of MBTCA and NaCl were prepared with molar
mixing ratios of MBTCA:NaCl = 1:1, 1:2, 1:3, and 2:1. Mono-/di-/tri-sodium MBTCA salt solutions were
obtained by mixing MBTCA and NaOH (>99.9% purity, Sigma-Aldrich) with molar ratios of
MBTCA:NaOH = 1:1, 1:2, and 1:3, respectively. A mixture solution of MBTCA and monosodium
MBTCA salt with a molar mixing ratio of 1:1 was prepared as well. Aerosol particles were generated by
nebulizing the solutions using a single jet atomizer (HCT4810) on the Si wafer substrates (MTI
Corporation, 99.999% purity). The size of the droplets examined at RH > 90% ranged from 1 to 15 µm.

**2.2 In situ Raman microspectrometry (RMS) for particles deposited on a Si wafer**
During the hygroscopic measurements, in situ RMS was performed under a controlled RH to observe
the hygroscopic behavior, structural changes, and chemical compositional variations of the aerosols
generated from the solutions. The apparatus consisted of three parts: (A) see-through impactor, (B) Raman
microscope/spectrometer, and (C) humidity-controlling system. The Si wafer substrate was mounted on
the impaction plate in the see-through impactor. A more detailed discussion of the impactor and humidity-
controlling system can be found elsewhere (Gupta et al., 2015). Briefly, the RH inside the impactor was
controlled by mixing dry and wet (saturated with water vapor) $N_2$ gases. The flow rates of total 4 L·min⁻
¹ of the dry and wet $N_2$ gases were controlled by two mass flow controllers to obtain the desired RH in
the range of ∼1−95%, which was monitored using a digital hygrometer (Testo 645). The digital
hygrometer was calibrated using a dew-point hygrometer (M2 Plus-RH, GE) to provide RH readings with
±0.5% reproducibility. The Raman spectra and optical images of the aerosol particles were recorded by
Labspec6 using a confocal Raman microspectrometer (XploRA, Horiba Jobin Yvon) equipped with a ×50,
0.5 numerical aperture objective (Olympus). An excitation laser with a wavelength of 532 nm and 6 mW
power was used, and the scattered Raman signals were detected at specific RHs during the hygroscopic
measurements using an air-cooled multichannel charge-coupled device (CCD) detector. The data
acquisition time for each measurement was 120 s. The spectral resolution was 1.8 cm⁻¹ using 1800 gr/mm.
The optical images were recorded continuously in RH = 1% steps with a size of 904×690 pixels during
the first dehydration (by decreasing RH from ~95 to ~1%), followed by the humidification (by increasing





RH from ~1 to ~95%) experiments using a top video camera assembled in the Raman instrument and
processed using an image analysis software (Matrox, Inspector v9.0). The changes in particle size with
the RH were monitored by measuring the particle 2-D area in the optical images to generate hygroscopic
curves. These curves are represented by the area ratio ($A/A_0$) as a function of RH, where the 2-D projected
aerosol area at a given RH ($A$) is divided by that at the end of the dehydration process ($A_0$) (Ahn et al.,
2010). All hygroscopic experiments were conducted at room temperature (T = 22±1℃). Aerosol particles
generated from a pure NaCl aqueous solution to check the accuracy of the system showed typical
hysteresis curves with deliquescence RH (DRH) = 75.5(±0.5)% and efflorescence RH (ERH) = 46.3–
47.6%, which are consistent with the theoretical and reported values.

**2.3 SEM/EDX X-ray mapping of effloresced particles deposited on Si wafer**
SEM/EDX X-ray mapping was performed for effloresced particles to determine the morphology and
spatial distribution of the chemical elements after the hygroscopicity measurements of individual particles
(Ahn et al., 2010; Gupta et al., 2015). The measurements were carried out using a Jeol JSM-6390 SEM
equipped with an Oxford Link SATW ultrathin window EDX detector. The resolution of the detector was
133 eV for Mn Kα X-rays. The X-ray spectra and elemental X-ray maps were recorded under the control
of Oxford INCA Energy software. A 10 kV accelerating voltage and 0.7 nA beam current were used, and
the typical measuring time for the elemental mapping was five minutes. An elemental quantification
procedure, which is well described elsewhere (Wu et al., 2019a), was used for obtaining the elemental
concentration.

**2.4 In situ RMS assembled with levitation system**
The levitation experimental set up consisted of coupling an acoustic (ultrasonic) levitator equipped
with an environmental cell to an RMS, as shown in Fig. S1. The theory of acoustic levitation is described
in detail elsewhere (Seaver et al., 1989). An ultrasonic levitator was modified (APOS BA 10, Tec5,
Germany) to be installed within an environmental levitation cell consisting of two quartz windows,
allowing the particle analysis (Seng et al., 2018). Two inlet/outlet valves were used for gas supplies to
modify the relative humidity (RH) inside the cell. A sensor (SHT75 Sensirion) was placed into the cell to





control the RH and temperature. The RH inside the chamber was controlled by mixing dry and wet Ar
gases with a flow rate of 200 mL·min$^{-1}$ in the range of 10-80% ($\pm$1%) RH, and the temperature was T =
25$\pm$3°C, making the experiments close to static flow conditions. The control of humidity and temperature
allows limited droplet evaporation and long-term monitoring of the particles. The RMS measurements
were performed with a LabRAM HR evolution confocal spectrometer (Horiba Scientific, S.A) at certain
RHs first during humidification and then during dehydration. The instrument was equipped with an ×50,
0.45 numerical aperture Olympus objective (WD = 13.8 mm) and a He-Ne laser ($\lambda$ = 632.8 nm – 6 mW)
with a theoretical lateral resolution of ~2 µm, and a depth of the laser focus corresponding to 16 µm with
a $\Delta z$ limit $\geq \pm 3$ µm. The cell was mounted on an XYZ stage under the objective, allowing an adjustment
of the droplet in the optimal position for the measurements. The mean size of the initial droplet injected
in the levitator was 80 µm. The Raman spectra and optical images recorded at specific RHs were analyzed
similarly to those obtained on the Si wafer.

**2.5 Measurement of acid dissociation constants of MBTCA**
MBTCA is a tri-carboxylic acid with three acid dissociation constants. To determine the three
constants, a 0.02 M, 5 ml MBTCA solution was titrated with a 0.1 M NaOH solution, where the constants
were determined based on the Henderson-Hasselbalch equations (Harris, 2012):
$H_3M + OH^- \rightarrow H_2M^- + H_2O$     $pH = pKa_1 + \log([H_2M^-]/[H_3M])$
$H_2M^- + OH^- \rightarrow HM^{2-} + H_2O$     $pH = pKa_2 + \log([HM^{2-}]/[H_2M^-])$
$HM^{2-} + OH^- \rightarrow M^{3-} + H_2O$     $pH = pKa_3 + \log([M^{3-}]/[HM^{2-}])$
where $H_3M$, $H_2M^-$, $HM^{2-}$, and $M^{3-}$ represent aqueous MBTCA, mono-, di-, and tri-sodium MBTCA anions,
respectively. The $pKa_1$, $pKa_2$, and $pKa_3$ are the pHs when $[H_3M]$, $[H_2M^-]$, and $[HM^{2-}]$ equal $[H_2M^-]$,
$[HM^{2-}]$, and $[M^{3-}]$, respectively, during the acid-base titration. Specifically, when NaOH was added at 0.5,
1.5, and 2.5 ml, the corresponding pHs of the solution are the three constants, which were 3.59, 4.85, and
6.79. Fig. 1 shows the calculated titration curve of MBTCA using the three determined Ka values, which
is the same as the experimentally obtained titration data, supporting the validity of the Ka values, which
were not reported so far.





## 3. Results and Discussion

### 3.1 Hygroscopic behavior of pure MBTCA particles

Wet-deposited MBTCA aerosols exhibited three types of hygroscopic behavior. As shown in Fig. 2, during the dehydration process, the exemplar droplets of types 1 and 2 shrank continuously with decreasing RH due to water evaporation until RHs = 58.4% and 40%, and then effloresced promptly at RH = 57.8% and gradually at RH = 39 - 35%, respectively. The effloresced particles maintained their size and shape with further decreases in RH. Whereas, the type 3 aerosols decreased continuously in size without a distinct change from RH = 94% to RH = 3% during the dehydration process. During the humidification process, types 1 and 2 particles kept the same size and shape until RH = ~90%, while type 3 particles experienced a phase change at RH = 36.7% and remained the same until RH = ~85%. Fig. 2 also presents the corresponding optical images and in situ Raman spectra to assess the structural evolution of the MBTCA aerosols during the dehydration and humidification processes. Briefly, Raman peaks at ~1411 - 1420 $cm^{-1}$, ~1460 and ~2950 $cm^{-1}$, ~1660 - 1730 $cm^{-1}$, and ~3420 - 3475 $cm^{-1}$ are for vibrations of C=O from COO⁻, CH, C=O from COOH, and OH from water, respectively (Edsall, 1937; An et al., 2016). The redshift of the C=O peak (from COOH) from 1715 to 1660 $cm^{-1}$ with decreasing FWHH (full width at half height), which is consistent with the standard MBTCA crystal, and the irregular shape and rough surface of types 1 and 2 aerosols at RH = 57.8% and 35%, respectively, confirmed that the particles effloresced into a solid phase. The optical images in the inset above the hygroscopic curve of the type 2 particles showed gradual efflorescence at RH = 39 − 35%. The water peak at ~3475 $cm^{-1}$ disappeared as well after the efflorescence. In contrast, type 3 aerosols maintained a circular morphology until RH = 3%, as shown in the optical images in Fig. 2, even though an overlapped C=O (from COOH) peak at 1660 - 1680 $cm^{-1}$ appeared during the dehydration process, and the water peak became undetectable, as shown in the Raman spectra at RHs = 45% and 3%, suggesting an amorphous/solid-state and the presence of an activation barrier or diffusional resistance to homogeneous nucleation required for the crystallization of MBTCA droplets as efflorescence is a kinetically controlled process (Martin, 2000; Freedman, 2017). Previous studies reported that α-pinene SOAs were very likely to exist as a highly viscous semisolid or even glassy state at low humidity (Saukko et al., 2012; Renbaum-Wolff et al., 2013; Berkemeier et al., 2014; Dette et al., 2014; Kidd et al., 2014; Song et al., 2016; Lessmeier et al., 2018). In addition, many



organic substances, such as carboxylic acids, carbohydrates, and proteins, tend to form amorphous rather
than crystalline phases upon the drying of aqueous solution droplets (Mikhailov et al., 2009). The different
behavior of the MBTCA particles can be attributed to different nucleation mechanisms, i.e., homogeneous
and heterogeneous nucleation, for pure and impure (seed-containing) MBTCA particles, respectively. A
similar situation was reported for $NH_4NO_3$, $NaNO_3$, and $NH_4HSO_4$ particles (Lightstone et al., 2000;
Hoffman et al., 2004; Gibson et al., 2006; Kim et al., 2012; Jing et al., 2018; Sun et al., 2018; Wu et al.,
2019b). The Si substrates used in this study could also facilitate efflorescence (Eom et al., 2014; Wang et
al., 2017). During the humidification process, the Raman spectra and morphology remained unchanged
for types 1 and 2 particles until RH = ~90%, where a slight change in morphology was observed due to
structural re-arrangements by the absorption of moisture on the lattice imperfections (Gysel et al., 2002).
Type 3 particles during the humidification process became irregular in shape, and the overlapped C=O
(from COOH) peak shifted to 1660 cm$^{-1}$ at RH = 36.7%, as shown in the optical image and Raman
spectrum, indicating the formation of solids. With the further increase in RH, particles maintained their
size and shape until RH = 85%, where they started to decrease in size due to a re-arrangement in structure.
The efflorescence of laboratory-generated particles during the humidification process was reported
previously in the $NaCl$-$MgCl_2$ mixture system as the condensed water can help overcome the kinetic
barrier, leading to crystallization (Gupta et al., 2015). All types of MBTCA particles maintained the
crystal phase until RH = 95%. Among 100 particles, type 1-3 particles accounted for approximately 25%,
5%, and 70%, respectively. Based on the experimental results, MBTCA droplets have DRH > 95% and
ERH = 30–58%. This is the first study reporting the hygroscopic properties of MBTCA. A previous study
showed that MBTCA was not hydrated significantly in the ambient atmosphere (Kildgaard et al., 2018),
implying that the MBTCA solids stay in the air once they effloresced, based on our results.

**3.2 Hygroscopic behavior of mono-/di-/tri-sodium MBTCA salt aerosols**
The hygroscopicity and Raman spectra of mono-/di-/tri-sodium MBTCA salt aerosols (hereafter,
denoted as $NaH_2M$, $Na_2HM$, and $Na_3M$, respectively) were studied to examine the hygroscopic behavior
and estimate the chemical reactivity of MBTCA with NaCl. Figs. 3(a)-(c) show the 2-D projected area
ratio plot of aerosol particles generated from 0.3 M $NaH_2M$, $Na_2HM$, and $Na_3M$ aqueous solutions as a





function of the RH together with the corresponding optical images and Raman spectra recorded at specific
RHs. As shown in Figs. 3(a) and (b), NaH$_2$M and Na$_2$HM aerosols shrank and grew continuously without
a phase transition during the dehydration and humidification processes, respectively, which is also
reflected in the optical images and Raman spectra, where they maintained their circular morphology only
with a change in size and the same Raman peak patterns and positions with small variations in the relative
peak intensities during the entire process. The water peak at ~3400-3500 cm$^{-1}$ can still be observed at the
end of the dehydration process. Even after being kept in a desiccator for two months, NaH$_2$M and Na$_2$HM
particles still showed the same shapes and Raman spectra with those at RHs = 3.4% and 2.8%,
respectively. These results indicate the non-crystallizable properties and supersaturated amorphous phase
state of the particles. The Na$_3$M particles behaved differently as they did not crystallize during the
dehydration process. On the other hand, the aerosols exhibited efflorescence at RH = 46.8% during the
humidification process (Fig. 3(c)), deliquesced to become a droplet at RH = 53.1%, and grew continuously
after that with increasing RH. The Raman spectra of the Na$_3$M particles in Fig. 3(c) showed that the peak
at 1420-1460 cm$^{-1}$ became two sharp peaks when the particles effloresced, and the OH peak at 3400 cm$^{-1}$
indicates that Na$_3$M particles possibly exist in the hydrated form. The Na$_3$M particles behaved
analogously to type 3 MBTCA particles, which might be due to their similar structures when all three
COOH in MBTCA were replaced with COONa upon the reaction between MBTCA with NaOH. Based
on the top Raman spectra of aqueous MBTCA, NaH$_2$M, Na$_2$HM, and Na$_3$M aerosols in Figs. 2 and 3, the
ratios of the CH peak at ~1460 cm$^{-1}$ to the C=O peak at ~1720 cm$^{-1}$ (from COOH) and to the C=O peak
at ~1420 cm$^{-1}$ (from COO$^-$) increased and decreased in the order of MBTCA, NaH$_2$M, Na$_2$HM, and Na$_3$M
because of their reduced and elevated levels of COOH and COO$^-$, respectively.

**3.3 Hygroscopic behavior of MBTCA-NaCl mixture aerosols**
Aerosols were generated by the nebulization of MBTCA-NaCl mixture solutions of molar mixing
ratios of MBTCA:NaCl = 1:1, 1:2, 1:3, and 2:1 and deposited on Si wafer substrates, while maintaining
the entire hygroscopic measurement system at RH > 90%. The hygroscopic behavior was investigated for
~10 individual aerosols of each mixing ratio, which are discussed in the following sections.





### 3.3.1 Aerosols generated from solutions of MBTCA:NaCl = 1:1 and 2:1

Fig. 4 presents the hygroscopic curves of representative aerosols nebulized from solutions of MBTCA:NaCl mixtures at different molar ratios (1:1 and 2:1) along with the corresponding optical images and Raman spectra at specific RHs. During the dehydration process, the circular liquid droplets decreased in size gradually without any noticeable phase change. The Raman peak patterns were maintained only with the C=O peak at 1721 cm$^{-1}$ (from COOH) shifting mildly rightwards, the water peak at 3466 cm$^{-1}$ becoming undetectable, and the relative peak intensities at ~1411, 1457, and 1721 cm$^{-1}$ varied when the RH was as low as 1.2%, indicating that the liquid droplets formed amorphous solids. The peak at 1680 cm$^{-1}$ on the Raman spectra of MBTCA:NaCl = 2:1 at RH = 1.2% suggested that the amorphous structure of the remaining MBTCA had been retained. Both MBTCA and NaCl have their DRHs and ERHs. Therefore, a step-wise efflorescence would happen if it is assumed that the mixture aerosols are an MBTCA-NaCl binary system, i.e., a component of the aqueous droplets precipitates first at their specific ERHs depending on their mixing ratios, and the second crystallization from the remnant eutonic solution occurs at their mutual ERH (MERH) with further decreases in RH, which is independent of the mixing ratios, generally forming a heterogeneous, core-shell crystal structure owing to the two-stage crystallization process (Ge et al., 1996; Gupta et al., 2015). However, the particles of MBTCA:NaCl = 1:1 and 2:1 mixing ratios did not follow the step-wise transitions in the present study, revealing that the aerosols do not belong to the MBTCA-NaCl binary system and the chemical compositions evolved during the hygroscopic experiment due to the reaction between MBTCA and NaCl, which will be discussed later.

During the humidification process, aerosol particles of two mixing ratios grew continuously when the RH was increased from 1.2% to 90% with C=O peak (from COOH) shifting back to ~1721 cm$^{-1}$ and the water peak becoming significant, as shown in Fig. 4. Several small crystal-like spots, which are marked by a dotted circle on the inset optical image beside the hygroscopic curve in Fig. 4(a), appeared in the particles with the mixing ratio of MBTCA:NaCl = 1:1 when the RH was increased to 67.2% and dissolved completely at RH = 71.2%. As the Raman spectra did not show any signals of the crystallized organics and RH = 71.2% is close to the DRH of pure NaCl (75($\pm$0.5)%), the crystal-like moieties should result from the effloresced NaCl. The more noticeable water peak in the Raman spectrum taken at RH = 71.2% than that at RH = 67.2% also supports that the NaCl dissolved at RH = 71.2% as NaCl is quite hygroscopic


(Li et al., 2017). No phase transition of NaCl was detected during the dehydration process, probably
because the supersaturated organic moiety inhibited the crystallization of NaCl. The observation of
effloresced particles during the humidification process might be caused by the structural re-arrangement
of the amorphous particles upon the slow and continuous absorption of moisture with increasing RH
(Mikhailov et al., 2009), leading to less restriction to NaCl crystallization. Indeed, organics in organic-
inorganic mixture aerosols were reported to be a minor disturbance to the DRH of inorganic salts; in
contrast, they may markedly decrease the ERH of inorganic salts depending on the organic type (Parsons
et al., 2004).

**3.3.2 Aerosols generated from solutions of MBTCA:NaCl = 1:2 and 1:3**

Fig. 5 shows the hygroscopic curves of aerosol particles nebulized from solutions of MBTCA:NaCl

with molar mixing ratios of 1:2 and 1:3, together with the corresponding optical images and Raman
spectra at the transition RHs. During the dehydration process, droplets from the solutions of
MBTCA:NaCl = 1:2 and 1:3 decreased gradually in size owing to water evaporation until a single-stage
transition was observed at RHs = 47.2-46.5% and 46.7-45.8%, respectively, where the particle shape
became less circular in the optical images. At this point, the following were observed in the Raman spectra:
the water peak at 3455 cm$^{-1}$ disappeared; the C=O peak at ~1722/1720 cm$^{-1}$ (from COOH) shifted slightly
rightwards; the relative peak intensities at 1417/1416, 1461, and 1722/1720 cm$^{-1}$ varied. With the further
decreases in RH until ~6%, the particles kept their size and shape. During the humidification process, all
particles of MBTCA:NaCl = 1:2 and 1:3 maintained their structure until RHs = 50% and 40%,
respectively, where they experienced a size decrease due to structural re-arrangement until RH = ~70%,
grew continuously to become circular at RH = ~73%, and totally deliquesced into homogeneous droplets
at RHs = 73.9% and 74.5%, respectively. Particle size and water peak increased rapidly, and the C=O
peak (COOH) shifted back to 1720 cm$^{-1}$. Upon a further increase in RH, they grew continuously by water
absorption. The ERH and DRH were attributed to the NaCl moiety as the Raman spectra maintained the
peak patterns during the entire process, and the organic components condensed onto the NaCl crystal core
almost simultaneously as an amorphous shell when efflorescence occurred, which is also indicated by the
optical images. Before the complete deliquescence of the NaCl crystal core, the water peak at ~3455 cm$^{-}$



[1] in the Raman spectra and the optical images at RH = 72.4% and 73.8% of the particles from the
MBTCA:NaCl = 1:2 and 1:3 solutions show that the organic shell was in the liquid phase, meaning that
the mixture particles were in a solid-liquid equilibrium state (Sun et al., 2018).

All the particles from MBTCA:NaCl = 1:2 and 1:3 solutions showed hysteresis curves with ERHs in

the range of 46.7-45.2% and 47.2-45.6%, respectively, and DRHs = 73.9($\pm$0.3)% and 74.5($\pm$0.3)%,
respectively.

### 3.3.3 Chemical reactivity of aerosols generated from MBTCA−NaCl mixture solutions

The first Raman spectra of the aerosols generated from MBTCA-NaCl mixture solutions in Figs. 4

and 5 were obtained before the dehydration process, which are comparable to that of pure MBTCA droplet
particle in Fig. 2 except for a much stronger free water peak at 3450-3470 cm$^{-1}$ due to the presence of a
more hygroscopic NaCl moiety. This suggests that upon nebulization from the solutions, the mixture
droplets were mostly the MBTCA-NaCl binary system. The Raman spectra obtained at the beginning of
the dehydration process and the end of the humidification process revealed increased and decreased ratios
of the CH peak at ~1460 cm$^{-1}$ to the C=O peaks at ~1720 cm$^{-1}$ (from COOH) and ~1412 cm$^{-1}$ (from
COO$^-$), respectively, which implies that the reaction between MBTCA and NaCl occurred during the
hygroscopic experiment, leading to the decreased and increased levels of the COOH and COO$^-$ moieties,
respectively. Fig. 6(a) presents the Raman spectra of particles generated from MBTCA:NaCl = 1:1, 1:2,
and 1:3 solutions together with that of NaH$_2$M particles obtained at the end of humidification by
normalizing to the CH peak at 1458 cm$^{-1}$. The C=O peak intensities at 1720 cm$^{-1}$ (from COOH) and 1412
cm$^{-1}$ (from COO$^-$) of the particles generated from the mixture solutions were higher and lower,
respectively, than those of the NaH$_2$M particle, suggesting that the aerosols generated from the MBTCA-
NaCl solutions produced only NaH$_2$M as the reaction product between MBTCA and NaCl, regardless of
the mixing ratios. The droplet particles after the humidification process were present as an MBTCA-
NaCl-NaH$_2$M ternary system with varying compositions. As the first acid dissociation constant of
MBTCA (pKa$_1$ = 3.59) is more than 1 and 3 orders of magnitude larger than the second (pKa$_2$ = 4.85) and
third (pKa$_3$ = 6.79), respectively, H$_2$M$^-$ is more abundant than HM$^{2-}$ and M$^{3-}$. The chemical reaction
between NaCl and MBTCA would occur in the aqueous phase as follows:




MBTCA(aq) + $H_2M^-$(aq) + $H^+$(aq)+ $Na^+$(aq) + $Cl^-$(aq) → MBTCA(aq) + $H_2M^-$(aq) + $Na^+$(aq) +
$Cl^-$(aq) + HCl(g)↑ → $NaH_2M$ (+ MBTCA, amorphous) + NaCl(s) after the efflorescence

The $NaH_2M$ particles may exist as amorphous particles, as described before in section 3.2. Raman spectra
of standard aerosols generated from solutions of MBTCA:$NaH_2M$ = 0:1, 1:1, and 1:0 were obtained at
different RHs to estimate the chemical reactivity of the aerosol particles generated from the MBTCA-
NaCl mixture solutions, which were used as a calibration curve to help determine the relative MBTCA
and $NaH_2M$ contents in the aerosols at specific RHs. The estimation of the chemical reactivity between
malonic acid and NaCl performed in the similar way was reported in a previous study (Li et al., 2017).
The Raman spectra of MBTCA, $NaH_2M$, and mixture aerosols of MBTCA:$NaH_2M$ = 1:1 obtained at RH
= 90% and normalized to the $CH_3$ peak at 1460 $cm^{-1}$ showed that the intensity ratio of the two peaks at
1460 $cm^{-1}$ ($CH_3$) and ~1720 $cm^{-1}$ (C=O from COOH) (i.e., $I_{1460}/I_{1720}$) increased with increasing $NaH_2M$
level because of the decreased COOH content, as shown in Fig. 6(b). The ratio, $I_{1460}/I_{1720}$, for each
standard aerosol exhibited good linearity as a function of RH, as shown in Fig. 7(a), where the mean
values obtained from 10 aerosols of each standard aerosol sample are plotted with error-bars. The Raman
intensity ratios of the standard aerosols increased with decreasing RH because the C=O stretching
vibrations of the free COOH group in the aqueous phase and the intramolecular hydrogen-bonded COOH
group in the supersaturated phase become weaker and stronger (Bertran et al., 2010), respectively, with
decreasing RH during the dehydration process.

The dependency of the $I_{1460}/I_{1720}$ ratios on RH can be used to estimate the MBTCA and $NaH_2M$

(monosodium MBTCA salt) contents in the NaCl-MBTCA aerosols at specific RHs based on the
calibration curve and to calculate the further reactivity. The chemical reactivity of the mixtures is
represented as the degree of the reaction, which is defined as the ratio of consumed to the original amount
of the limiting reactant. For example, for aerosols from solutions of MBTCA:NaCl = 2:1 and 1:2, NaCl
and MBTCA are the limiting reactants, respectively. Fig. 7(b) shows the degree of the reaction of aerosols
generated from solutions of each mixing ratio, where the mean degree of reaction has ~1.5-4% deviations
owing to statistical variations in the Raman peak intensities caused by the baseline correction procedure





and the uncertainties involved in the calibration measurements. The reactivity was estimated at five stages
during one cycle hygroscopic experiment.

*Stage 1;* At the beginning of the hygroscopic experiment, no reaction occurred for all the mixed

droplets based on their Raman spectra, i.e., the degree of the reaction is 0.


*Stage 2;* As the RH decreased during the dehydration process, the reaction continued in the

aqueous aerosols until efflorescence of the droplets with mixing ratios of MBTCA:NaCl = 1:2 and

1:3 had occurred, and until the water content of the aerosols with mixing ratios of MBTCA:NaCl

= 1:1 and 2:1 became insignificant. The degrees of the reaction of aerosols with mixing ratios of

1:1, 1:2, and 1:3 were approximately 30%, whereas that of 2:1 approached 85%.

*Stage 3;* The reaction of aerosols generated from the solution of mixing ratio of MBTCA:NaCl =

2:1 was complete at the end of the dehydration process, indicating the total consumption of NaCl

and the formation of an MBTCA:NaH$_2$M = 1:1 mixture aerosol. The Raman spectra of the aerosols

with mixing ratios of MBTCA:NaCl = 1:1, 1:2, and 1:3 at the end of the dehydration process were

unsuitable for the reactivity estimation mostly due to their heterogeneous structure in the presence

of a NaCl core.

*Stages 4 and 5;* The reaction proceeded after deliquescence when the free H$^+$ and Cl$^-$ became

available again for aerosols with mixing ratios of MBTCA:NaCl = 1:1, 1:2, and 1:3, and a small

increase in the degree of reaction (~5%) was observed at the end of humidification for these

mixture droplets.


Most of the reactions occurred in the aqueous phase during the dehydration process with considerable
amounts of aqueous H$^+$ from MBTCA and Cl$^-$ from NaCl available for HCl liberation. During the entire
experiment, the reactivity followed the sequence of MBTCA:NaCl = 2:1 > 1:3 > 1:2 > 1:1, where the
reactivity appeared to be enhanced when either of the reactants is enriched. On the other hand, the reaction





was complete only when aqueous $H^+$ was sufficiently available, i.e., the reaction depended mostly on the
triacid level. The real-time aerosol mixture components based on the reactivity estimation of each mixing
ratio at specific RHs are shown on the hygroscopic curves in Figs. 4 and 5.

The morphology and elemental distribution of effloresced MBTCA-NaCl particles were examined by

SEM/EDX. Figure 8(a) shows the secondary electron images (SEIs) of the exemplar particles of each
mixing ratio. The elemental X-ray maps for MBTCA:NaCl = 1:1 and 1:2 particles suggest that the NaCl
solid moiety (represented by Na and Cl X-ray maps) crystallized homogeneously at small spots inside the
organic moiety. For MBTCA:NaCl = 1:3 particles with a significant amount of NaCl, the NaCl solid
existed as a core surrounded by the organic moiety. The organic mixture of MBTCA and $NaH_2M$
(represented by C and Na) condensed onto the NaCl core almost simultaneously when efflorescence
occurred, while maintaining a relatively circular morphology, even after being inserted into the vacuum
SEM chamber, which also indicates the low crystallization tendency of the organic moiety. The different
shapes of organic shell-inorganic core structures depending on the organic mass fraction and RH are
reported elsewhere (Karadima et al., 2019). The homogeneous structure of C and Na and the absence of
Cl for particles with mixing ratios of MBTCA:NaCl = 2:1, as shown in the corresponding SEIs and X-
ray spectrum in Figs. 8(a) and (b), confirmed that the reaction was complete at the end of the dehydration
process. The reaction between MBTCA and NaCl and the changes in the microstructures after the reaction
are expected to have some atmospheric implications since they may have enhanced ability to facilitate
further heterogeneous reactions in the atmosphere because of their low crystallization property. Na (from
both $NaH_2M$ and NaCl) and Cl (from NaCl) levels were used to estimate the degrees of reaction for the
MBTCA:NaCl = 1:1, 1:2, 1:3, and 2:1 systems, which were estimated to be ~25%, ~30%, ~37%, and
100%, respectively, with well matching to those from the Raman analysis by 5-8% differences.

**3.4 Hygroscopic behavior of pure MBTCA and MBTCA−NaCl mixture particles in the levitation**
**system**

The data acquired from the levitation system for contactless experiments on particles of ~80-100 µm

were used to compare with those obtained for aerosols on the Si wafer in the see-through impactor.



Two types of hygroscopic behavior of pure MBTCA particles were observed, corresponding closely
to types 1 and 3 aerosol particles in the see-through impactor system. In addition, the ERH was ~49-54%,
confirming that once overcome the kinetic barrier and effloresce into solids, the MBTCA particles no
longer capture water significantly. The Raman spectra and optical images are not shown separately.
The droplets composed of MBTCA:NaCl = 1:1, 1:2, and 1:3 mixing ratios were introduced into the
levitator and dried rapidly at RH = ~10% within 15 minutes (first rapid dehydration, i.e., the quenching
process), and humidified progressively to RH = 80%. Once RH = 80%, the particles dehydrated gradually
until RH = ~10% (second dehydration). The Raman spectra and optical images are shown in Fig. S2.
After the first rapid dehydration of the particles, the existence of peaks at 1660 and 1720 cm$^{-1}$ was
observed for all the mixtures, and the relative intensity of the peak at ~1720 cm$^{-1}$ increased with increasing
NaCl content, suggesting the formation of the mixture of solid MBTCA and amorphous moiety either
from MBTCA or NaH$_2$M. During the humidification process, the Raman peak at 1720 cm$^{-1}$ and the
particle size grew continuously with increasing RH. Transitions were observed at RH = ~71%, ~74.5%,
and ~75% for MBTCA:NaCl = 1:1, 1:2, and 1:3 mixture particles, respectively, with the water peak at
~3500 cm$^{-1}$ becoming significant for the three compositions. The observed transition points were
attributed to the deliquescence of NaCl within the particle with the MBTCA moiety partially remaining
as a solid phase, and the elevated NaCl content strongly enhanced the ability of the particles to uptake
water. The peak related to the solid portion at 1655 cm$^{-1}$ disappeared only for the MBTCA:NaCl = 1:3
mixture particles at the end of humidification, suggesting that the particle had transformed completely
into a droplet. During the second dehydration process, the particles showed the entire release of water, as
illustrated by the disappearance of the peak at 3500 cm$^{-1}$ at RH = ~50%, i.e., the ERH, for all the mixtures
while maintaining the peak patterns and positions until the lowest RH. The Raman spectra recorded at the
end of dehydration revealed both solid and amorphous phases for the MBTCA:NaCl = 1:1 and 1:2
mixtures due to the existence of the peaks at 1660 and 1720 cm$^{-1}$. In contrast, only the 1720 cm$^{-1}$
associated with the amorphous composition was observed for the MBTCA:NaCl = 1:3 mixture,
suggesting that the reaction between MBTCA and NaCl was facilitated extensively by the increased NaCl
concentration while absorbing sufficient moisture. The conspicuous DRHs and ERHs of all the mixtures
in the levitation system demonstrated a smaller degree of reaction between MBTCA and NaCl compared





to those obtained in the see-through impactor, which might be caused by the relatively closed atmosphere
in the levitator, i.e., less release of HCl, and the starting point of the hygroscopic cycle, i.e., the quenching
process resulting in partially effloresced MBTCA before the humidification process.

**4. Conclusions and atmospheric implication**
The hygroscopic behavior, physical states, and chemical reactivity of pure MBTCA particles, mono-
/di-/tri-sodium MBTCA salt particles, and MBTCA-NaCl particles of different mixing ratios were
examined by in situ RMS assembled with a see-through impactor as the starting point with dehydration.
The DRHs and ERHs of the laboratory-generated particles in the micrometer size range at room
temperature were determined by monitoring the change in the particle area in the 2-D optical images and
the corresponding Raman spectra at transition points with RH variation of ~1-95%. Pure MBTCA showed
three types of hygroscopic behaviors in that types 1 and 2 particles effloresced suddenly and gradually,
respectively, at ERH = 30-58% during the dehydration process, whereas type 3 particles crystallized
during the humidification process at RH = ~37%, not during the dehydration process because of a kinetic
barrier to nucleation with limited condensed water. Subsequently, all particles maintained their crystal
structure until RH = 95%. The mono- and di-sodium MBTCA salt aerosols did not show a clear
efflorescence RH (ERH) and deliquescence RH (DRH) during the dehydration and humidification
processes, respectively. In contrast, the tri-sodium MBTCA showed ERH = ~44.4-46.8% (during
humidification) and DRH = ~53.1%. The MBTCA-NaCl droplets with molar mixing ratios of
MBTCA:NaCl = 1:1 and 2:1 showed no distinct DRH and ERH because of the partial and complete
reactions with NaCl, respectively, whereas those with ratios of MBTCA:NaCl =1:2 and 1:3 experienced
single-stage efflorescence and deliquescence governed by the excess NaCl. Only monosodium MBTCA
($NaH_2M$) could be formed as a result of the reaction between NaCl and MBTCA regardless of the mixing
ratios, mostly during the dehydration process within the timescale of one to two hours according to Raman
analysis, indicating that the MBTCA-NaCl mixture systems are in an MBTCA-$NaH_2M$-NaCl ternary
system except when NaCl has reacted completely in the mixture aerosols of MBTCA:NaCl = 2:1 ratio.
The MBTCA-$NaH_2M$ existed as amorphous solids, even when the excess crystalline NaCl acted as a
heterogeneous nucleation core, which was also confirmed by X-ray mapping. The reaction occurred more



rapidly with a more elevated concentration of either MBTCA or NaCl, and the controlling factor for the
reactivity of the mixtures depended mostly on the availability of $H^+$ dissociated from the MBTCA
tricarboxylic acid. The hygroscopic experiments for pure MBTCA and MBTCA-NaCl mixture particles
were also performed in a levitation system with the starting point from humidification after the quenching
process and the RH variation of ~10 to 80%. The results acquired from the levitation system are consistent
with those obtained from the see-through impactor, only with less reaction between MBTCA and NaCl
resulting from the airtight atmosphere inside the levitator and the partial solidification of MBTCA after
the quenching process. In addition, the elevated NaCl moiety can eventually transform the solidified
MBTCA into droplets through reactions when absorbing adequate moisture.

These observations are expected to have important atmospheric implications in that they may help to

better understand the complexity of real ambient SOA and inorganic mixture particles. In this study, the
hygroscopicity of MBTCA was altered significantly when mixed with NaCl due to the reaction, so that
they are more likely to contribute to further gas-particle interactions. The amorphous phase state may
influence the uptake of gaseous photo-oxidants as well as the chemical transformation and aging of
atmospheric aerosols (Mikhailov et al., 2009). The observed aqueous shell with the solid core upon the
humidification of the mixture particles with mixing ratios of MBTCA:NaCl = 1:2 and 1:3 before the total
dissolution of NaCl can scatter solar radiation more efficiently (Adachi et al., 2011; Sun et al., 2018). The
aerosol liquid water can promote heterogeneous aqueous-phase chemical processes, resulting in the facile
formation of secondary aerosols (Cheng et al., 2016; Li et al., 2019). Recently, heterogeneous reactions
in aerosol water were reported to be a significant mechanism for haze formation in North China (Sun et
al., 2018). Overall, the hygroscopic curve, Raman signatures, and X-ray maps of the effloresced particles
provided clear features of the hygroscopic behavior and chemical reactivity of the MBTCA-NaCl mixture
system covered in this study. These results are expected to provide insights into the physicochemical
characteristics and atmospheric chemistry of highly oxidized SOAs mixed with inorganic particles.

*Data availability.* The data used in this study are available upon request; please contact Chul-Un Ro
(curo@inha.ac.kr).



*Author contributions.* LW, CB, SS, and CR designed the experiment. LW, CB, and SS carried out the
measurements and/or analyzed the data. LW, CB, SS, PF, EP, EV, YS and CR contributed discussion of
the data. LW, SS, and CR drafted the paper.

*Competing interests.* The authors declare that they have no conflict of interest.

*Acknowledgments.* This study was supported by Basic Science Research Programs through the National
Research Foundation of Korea (NRF) funded by the Ministry of Education, Science, and Technology
(NRF-2018R1A2A1A05023254). Authors thank the Region Nouvelle Aquitaine for the financial support
of the SPECAERO project. This work was performed through international and collaborative programs
supported by PHC STAR n° 38815XE and visiting scholars program from IDEX of the University of
Bordeaux.

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





Figure 1. Calculated titration curve for MBTCA, noted as $H_3M$ in this figure. The experimental data are
shown as orange triangles. 5 mL of 0.02 M $H_3M$ was titrated with a 0.1 M NaOH solution.

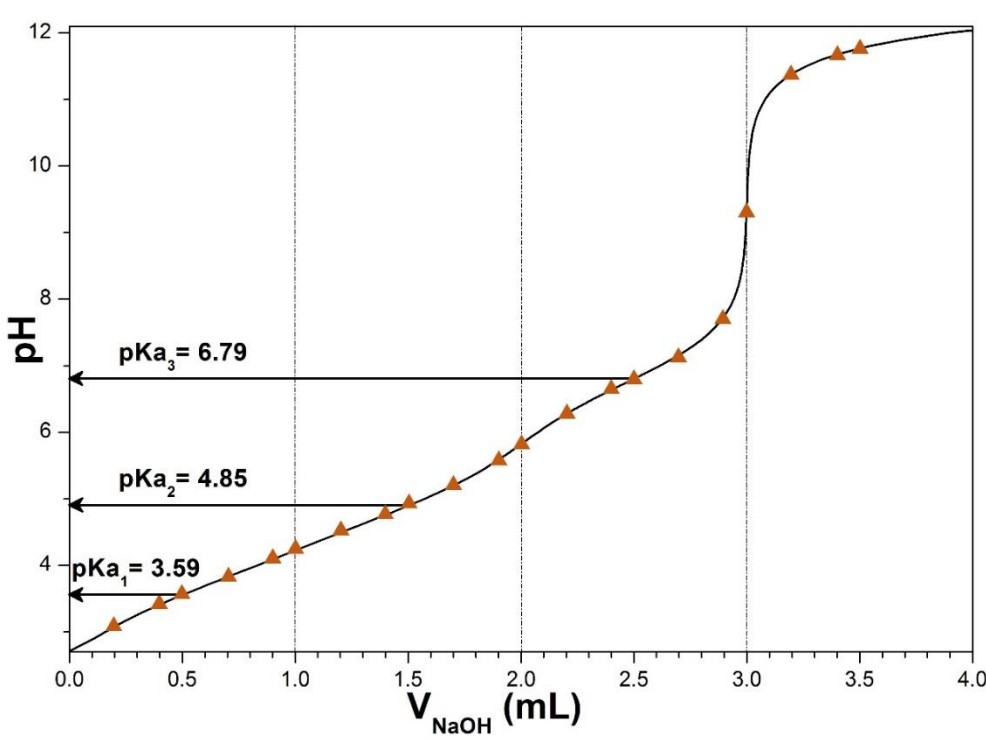



Figure 2. Hygroscopic curves, corresponding optical images, and Raman spectra at specific RHs of three
types of pure MBTCA particles. The transition RHs recorded during the dehydration (D) and
humidification (H) processes are marked with arrows in the hygroscopic curves.

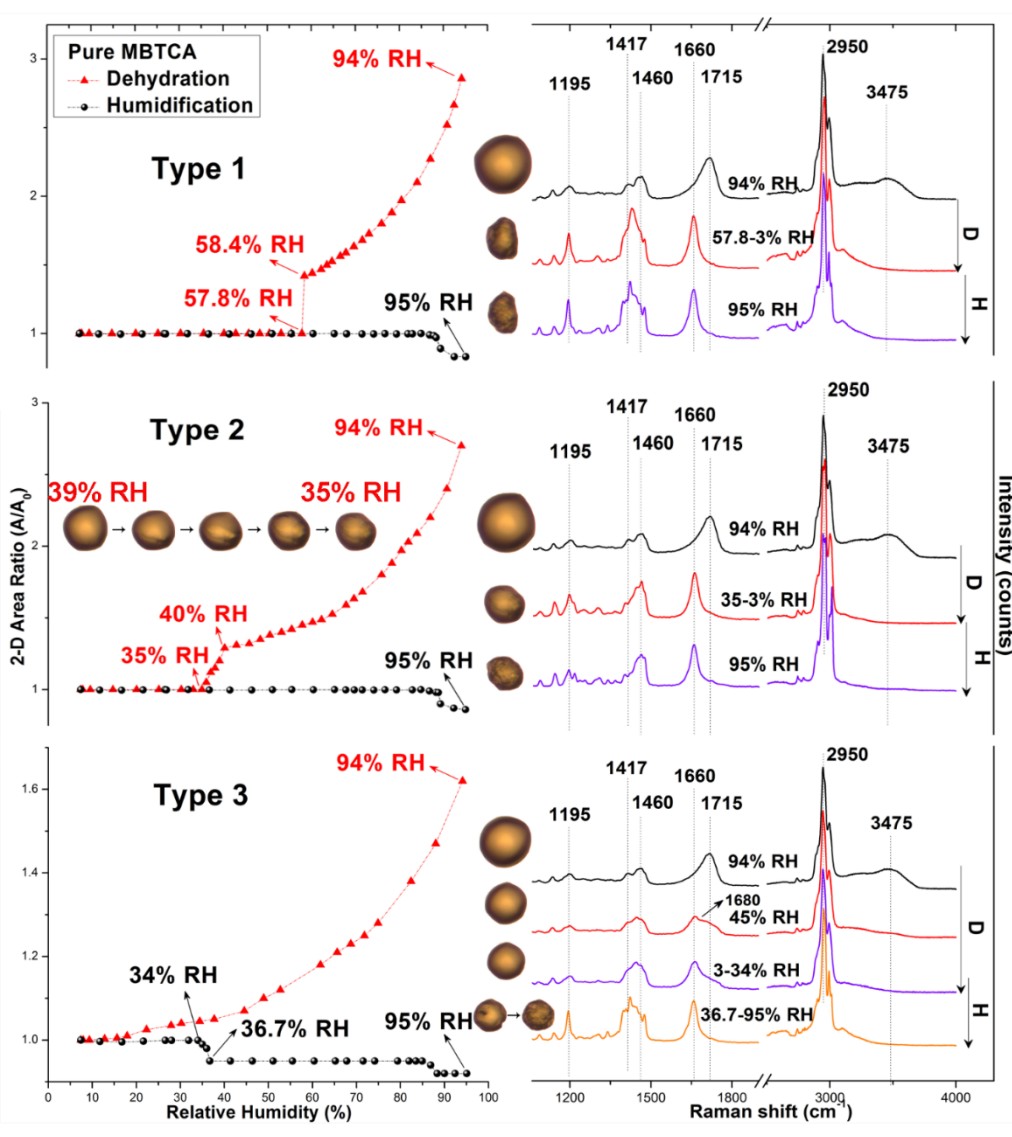

Figure 3. Hygroscopic curves, corresponding optical images, and Raman spectra at specific RHs of (a) mono-, (b) di-, and (c) tri-sodium MBTCA salt aerosols. The recorded transition RHs during the dehydration and humidification processes are marked with arrows in the hygroscopic curves.

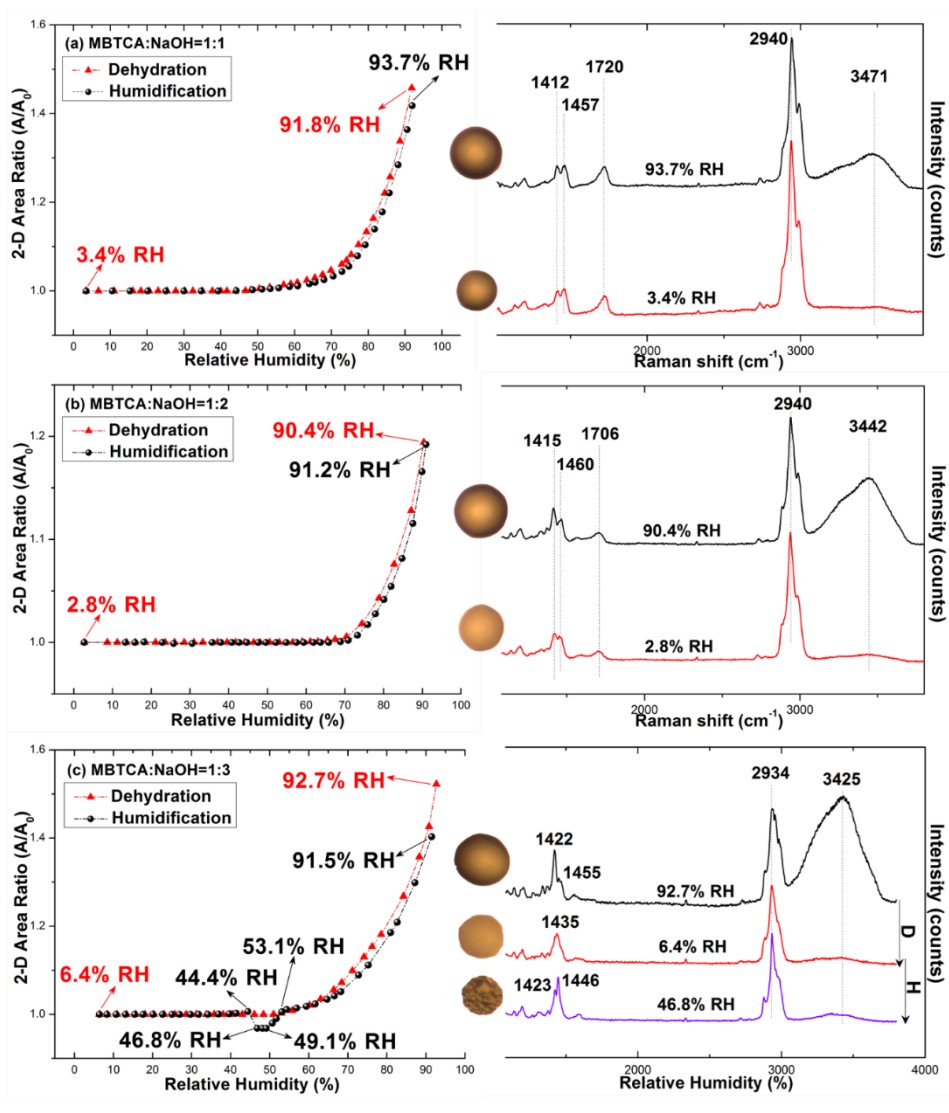

Figure 4. Hygroscopic curves, corresponding optical images, and Raman spectra at specific RHs of MBTCA:NaCl = (a) 1:1 and (b) 2:1.
The recorded transition RHs during the dehydration (D) and humidification (H) processes and the chemical compositions of the mixtures
at certain RHs are marked with arrows in the hygroscopic curves. The phase notations shown in parenthesis are s=solid; aq=aqueous;
and as=amorphous solid.

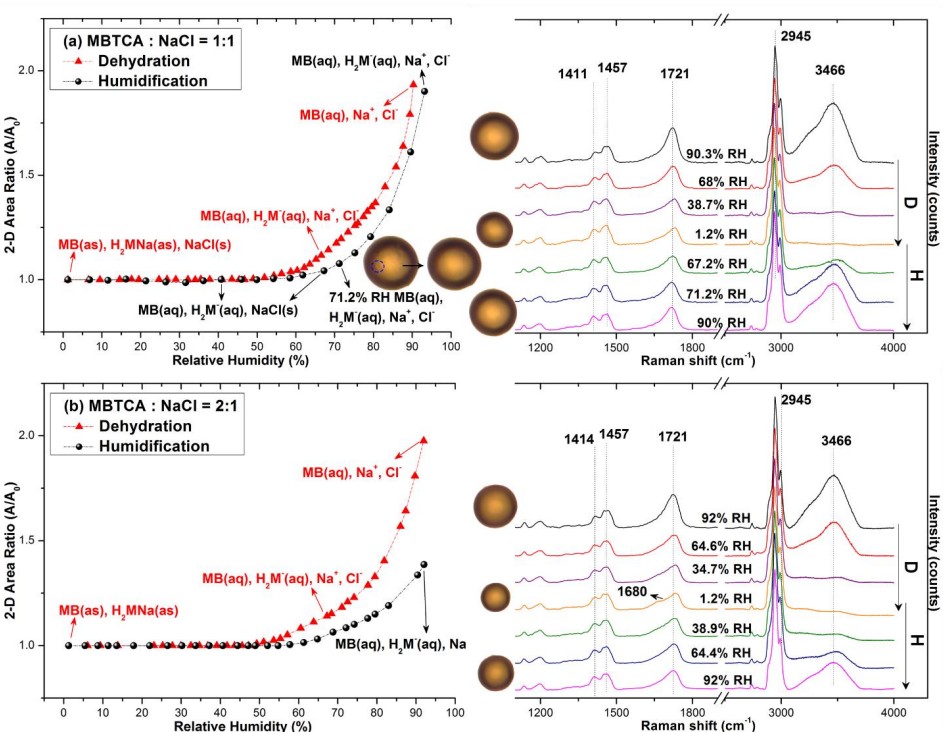



Figure 5. Hygroscopic curves, corresponding optical images, and Raman spectra at specific RHs of MBTCA:NaCl = (a) 1:2 and (b) 1:3. The recorded transition RHs during the dehydration (D) and humidification (H) processes and the chemical compositions of the mixtures at certain RHs are marked with arrows in the hygroscopic curves. The phase notations shown in parenthesis are s=solid; aq=aqueous; and as=amorphous solid.

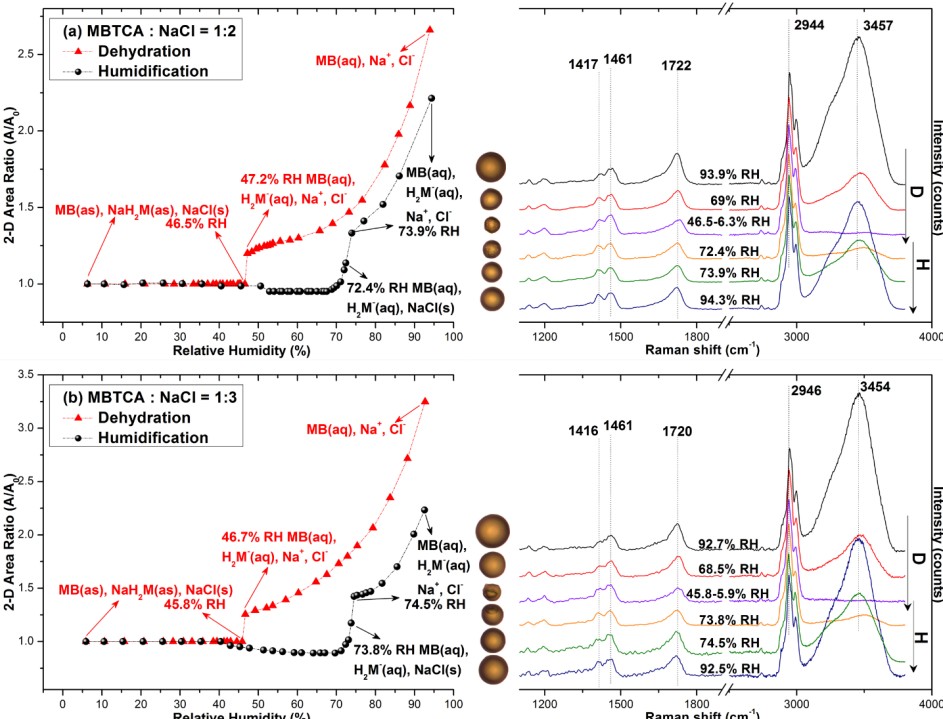



Figure 6. (a) Raman spectra of pure NaH$_2$M and mixture aerosols with mixing ratios of MBTCA:NaCl =
1:1, 1:2, and 1:3 obtained at the end of the humidification process, which were normalized to the CH$_3$
peak at 1458 cm$^{-1}$ and (b) Raman spectra of pure MBTCA, mixture of MBTCA:NaH$_2$M = 1:1, and pure
NaH$_2$M, which are normalized to the CH$_3$ peak at 1460 cm$^{-1}$.

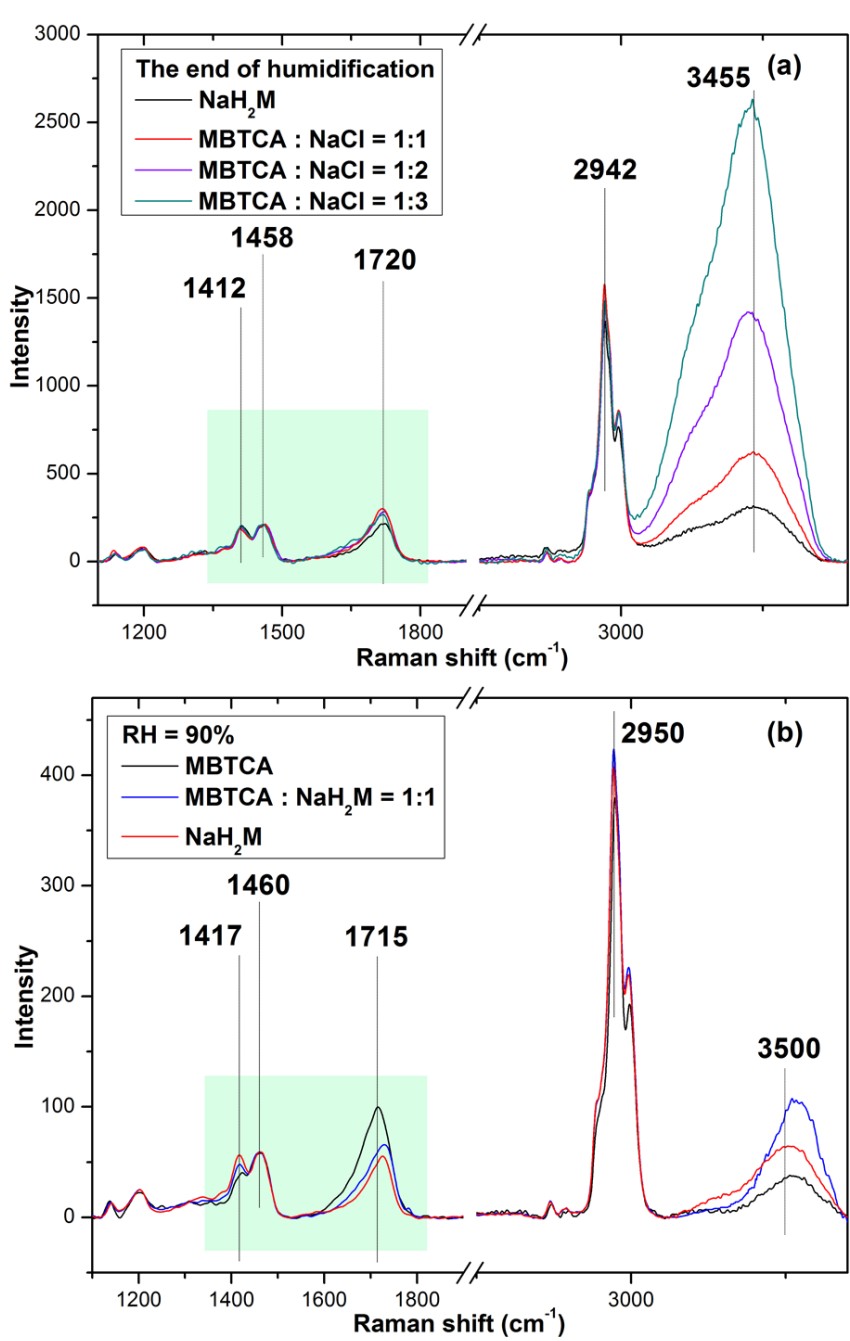





Figure 7. (a) Calibration curve calculated from the intensity ratios of two peaks at 1460 and 1720 cm$^{-1}$ as
a function of RH for NaH$_2$M, MBTCA:NaH$_2$M = 1:1, and MBTCA aerosols; (b) chemical reactivity
represented as the degree of reaction for mixture aerosols of MBTCA:NaCl = 1:1, 1:2, 1:3, and 2:1 during
the dehydration and humidification processes.

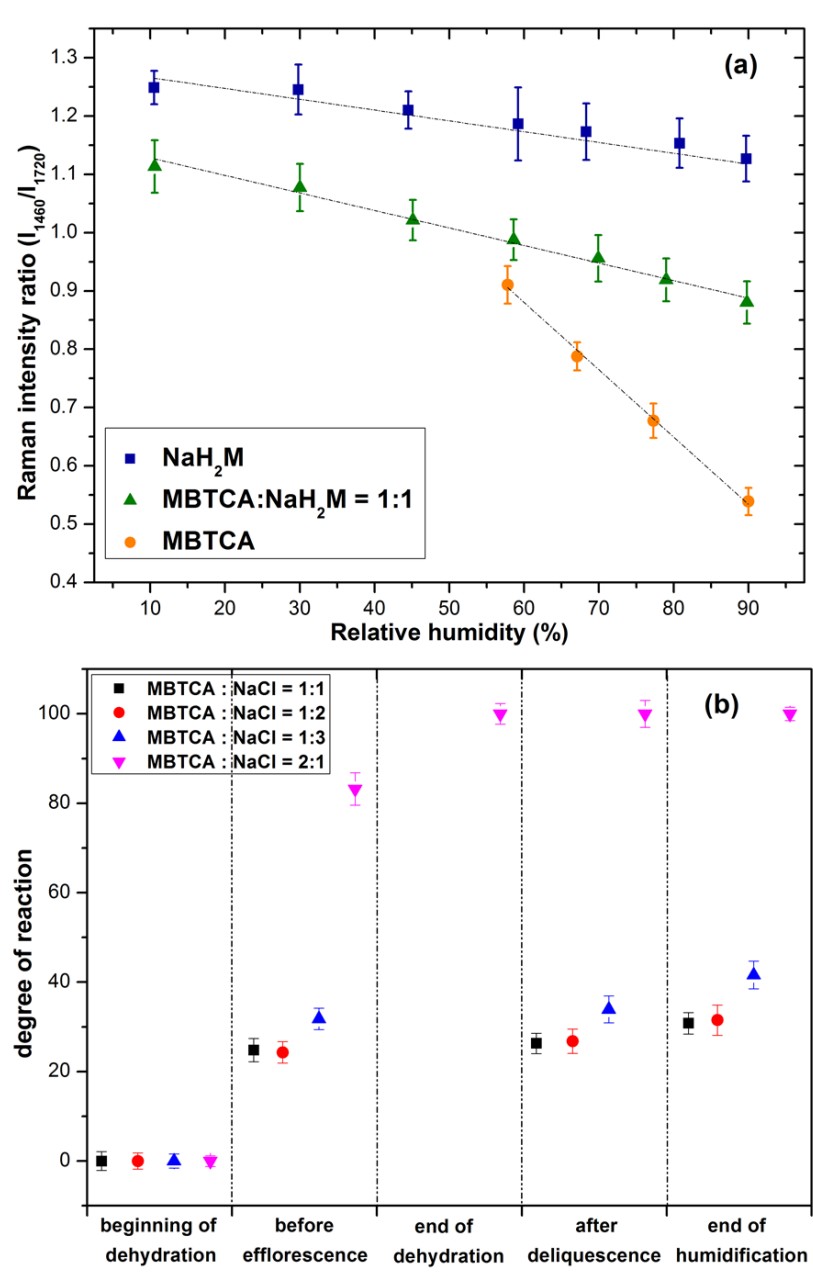





Figure 8. (a) Secondary electron images (SEIs) and elemental X-ray maps for C (from MBTCA and NaH₂M), Na (from NaH₂M and NaCl), and Cl (from NaCl). The scale bars are for 5 µm; (b) X-ray spectra and elemental concentrations of particles with four mixing ratios.

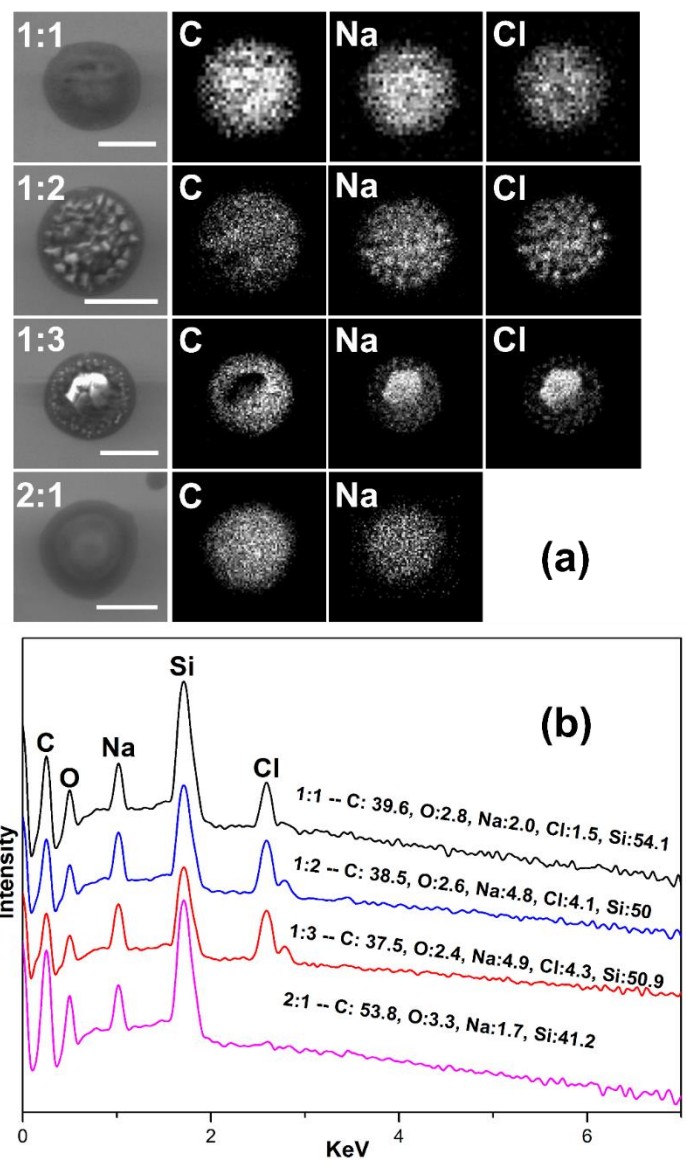