# Peer review of "Hygroscopic behavior of aerosols generated from solutions of 3-methyl-1,2,3-butanetricarboxylic"

_Atmospheric Chemistry and Physics, 2020_

## Referee Comment (RC1) · Anonymous Referee #1 · 26 Jun 2020

Li et al. report for the first time the hygroscopic measurements of MBTCA-inorganic system using Raman spectroscopy. The reactions involving HCl(g) release were discussed using the speciation of COOH and COO-, and related to the hygroscopic curves obtained by the optical imaging of the nominal particle size. This work would be an important increment to our knowledge of SOA hygroscopicity, but the current version is not ready for publication in ACP because of a number of issues as shown below.

The abstract is rather long. I suggest the authors to make the text more concise by focusing more results and their interpretation.

Lines 82-83: It is not clear how MBTCA can accelerate the new particle formation. The authors mentioned that MBTCA is extremely low volatile and one of the SOA components, i.e., one of the particulate compositions. Are you saying that preexisting aerosols enhance the new particle formation?

Line 127: I don't agree with the author's statement that RMS can provide the chemical compositions. RMS detects chemical functional groups of a molecule of interest.

Lines 170-173: It is useful if the authors show the experimentally measured hygroscopic curves of NaCl together with the thermodynamic model prediction.

Why were deposited and levitated particles examined in this study. The authors need to elaborate the necessity of using levitated particles.

In Figure 2, the authors need to explicitly define the type of MBTCA particles. How do types 1-3 of the particles differ?

Line 230: Please provide the evidence of the phase change at RH = 36.7%. The optical images of type 3 particle in Figure 2 show the morphology change from 34% RH to 36.7% RH, but it does not necessarily mean phase change.

Lines 250-252: Please clarify if the authors purposely injected impurity in the MBTCA particles. This is related to my earlier question: what is the type?

Line 270: It is confusing. The implication is based on the previous study or the current study?

In Figure 3, one of the interesting results is the efflorescence at 46.8% RH during the humidification. The explanation for this phenomenon is rather dry. The authors need to elaborate more. It would be also useful if the authors can discuss the repeatability of such a phenomenon.

Based on the chemical reactions the current study has proposed, acidic conditions are preferable for the reactions to proceed. However, Figure 7b shows the higher degree of

reaction for particles with MBTCA:NaCl = 1:3 than for those with that = 1:1. Particles with the higher abundance of NaCl (MBTCA:NaCl=1:3) are supposed to be less acidic. Do you have any explanation on it?

Figure 8a exhibits some heterogeneity of elemental distributions for particles with 1:2 and 1:3, which makes me wonder where Raman spectra were obtained. Raman spectra obtained from the edge and the center of the particles would be different.

The levitated particles show the lower reactivity than the deposited particles. The authors ascribed it to less release of HCl and the quenching process, which are too speculative. One more factor I want to point out is the almost one order of magnitude difference of the size between the two particles. It may not affect the reaction rates at a given RH, but it may do the release of HCl. I suggest the authors to discuss the potential size effect on the reactivity difference.

Interestingly, when the authors used the levitated particles, no size decrease (i.e., $A/A0 < 1$) was observed. The authors need to explain it. The size decrease observed with the deposited particles was attributed to the structural rearrange in the main text, but we may be seeing the effect of substrate.

---

## Author Comment (AC1) · 4 Jul 2020

**General Comment from Anonymous Referee #1**

Li et al. report for the first time the hygroscopic measurements of MBTCA-inorganic system using Raman spectroscopy. The reactions involving HCl(g) release were discussed using the speciation of COOH and COO-, and related to the hygroscopic curves obtained by the optical imaging of the nominal particle size. This work would be an important increment to our knowledge of SOA hygroscopicity, but the current version is not ready for publication in ACP because of a number of issues as shown below.

**Response:** We thank the reviewer very much for the careful evaluation and valuable comments for our work. Here we provide the response to the reviewer's comments. The final revision will be made later based on two reviewers' comments.

**Specific comments from Anonymous Referee #1 (comments are in italic)**

\* The abstract is rather long. I suggest the authors to make the text more concise by focusing more results and their interpretation.

**Response:** The abstract will be revised in a more concise way as much as possible respecting the reviewer's comment.

\* Lines 82-83: It is not clear how MBTCA can accelerate the new particle formation. The authors mentioned that MBTCA is extremely low volatile and one of the SOA components, i.e., one of the particulate compositions. Are you saying that preexisting aerosols enhance the new particle formation?

**Response:** As described in the papers (Donahue et al., 2013; Elm, 2019), MBTCA can nucleate the sulfuric acid vapor or provide polar functional groups to form and grow molecular clusters through hydrogen bonds, which promote the new particle formation. However, as this sentence is not directly related to our work, it would be omitted without deteriorating the context.

\* Line 127: I don't agree with the author's statement that RMS can provide the chemical compositions. RMS detects chemical functional groups of a molecule of interest.

**Response:** The reviewer is correct, and the sentence will be modified as "RMS can provide information on chemical functional groups, water contents, molecular interactions, and phase states of the aerosol particles".

\* Lines 170-173: It is useful if the authors show the experimentally measured hygroscopic curves of NaCl together with the thermodynamic model prediction.

**Response:** The hygroscopic curve of NaCl (as shown below) will be put into the supporting information as Fig S2.

\* Why were deposited and levitated particles examined in this study. The authors need to elaborate the necessity of using levitated particles.

**Response:** The deposited particles (~6.5  $\mu$ m in average in this study) may have some influences from the collecting substrate such as a facilitated heterogeneous nucleation, which can be eliminated in the levitation system due to the substrate-free and contactless properties. However, the particles in the levitation system are generally large in size (~80  $\mu$ m in average in this study), which is less atmospherically relevant. And thus the analysis of the particles in both systems was expected to give more detailed information on the hygroscopic behavior of MBTCA aerosols. The information will be elaborated in the revised text.

\* In Figure 2, the authors need to explicitly define the type of MBTCA particles. How do types 1-3 of the particles differ?

**Response:** The types of the particles were classified as, "type 1: with prompt efflorescence around ~50% RH during dehydration; type 2: with gradual efflorescence around ~35% RH during dehydration; type 3: with gradual efflorescence around ~37% RH during humidification", based on their different behavior when efflorescence occurred.

The different efflorescence behavior was attributed to different nucleation mechanisms: heterogeneous nucleation for types 1 and 2 particles, and homogeneous nucleation for type 3 particles.

MBTCA powders (98% purity, Toronto Research Chemicals, TCR), which was used for making the MBTCA solution, has intrinsic unknown impurity of 2%, and they were used without any purification. When MBTCA powders were dissolved in ultrapure de-ionized (DI) water (18 M $\Omega$ , Millipore Direct-QTM) and particles were generated by the nebulization of the aqueous solution using N2 gas (99.999% purity), impurities were either absent or associated with the droplets.

The impurities existed in types 1 and 2 particles after nebulization, acting as seed crystals to induce efflorescence. Aqueous moieties in particles were reported to effloresce more easily by heterogeneous nucleation in the presence of seeds (Schlenker and Martin, 2005; Li et al., 2014; Gupta et al., 2015). The lower ERH and gradual efflorescence of type 2 compared to type 1 particles might be due to the less amount of impurities. Type 3 particles contain negligible or no seed crystals, and large kinetic barrier and/or diffusional resistance make the formation of the crystal structure difficult owing to the decreasing availability of condensed water during dehydration, so that they did not experience any efflorescence. As type 3 particles account for 70% of all the analyzed one, it seems that MBTCA has slow homogeneous nucleation rate. The information will be explicitly described in the revised text.

**\* Line 230: Please provide the evidence of the phase change at RH = 36.7%. The optical images of type 3 particle in Figure 2 show the morphology change from 34% RH to 36.7% RH, but it does not necessarily mean phase change.**

**Response:** As shown in the optical image and Raman spectrum in Fig. 2, type 3 particles became irregular in shape during the humidification process, and the overlapped C=O (from COOH) peak at 1660 - 1680 cm-1 entirely shifted to 1660 cm-1 when the RH was increased from 34% to 36.7%, which indicates the phase change from amorphous/solid-state to solid state. Please also see the content at line 259-261 in the discussion paper, where the explanation has been done. The sentence at line 230 will be modified as, "type 3 particles experienced a sudden morphological change at RH = 36.7% and remained the same until RH = ~85%", for precise expression.

\* Lines 250-252: Please clarify if the authors purposely injected impurity in the *MBTCA* particles. This is related to my earlier question: what is the type?

**Response:** The MBTCA powders with intrinsic 2% impurity were used without any purification and no impurity was injected purposely. The information will be clarified in the revised text. Regarding the types of the particles, please see the response above.

\* Line 270: It is confusing. The implication is based on the previous study or the current study?

**Response:** The sentence will be modified as, "A previous study showed that MBTCA was not hydrated significantly in the ambient atmosphere (Kildgaard et al., 2018), and our results also implied that the MBTCA stay in the air as solids once they effloresced.", to avoid the confusion.

\* In Figure 3, one of the interesting results is the efflorescence at 46.8% RH during the humidification. The explanation for this phenomenon is rather dry. The authors need to elaborate more. It would be also useful if the authors can discuss the repeatability of such a phenomenon.

**Response:** Efflorescence during humidification was previously reported for Amazonian rain forest aerosols (Pöhlker et al., 2014) and the laboratory-generated NaCl–MgCl2 mixture particles (Gupta et al., 2015). And thus this phenomenon is not rare and it was claimed that the aerosol particles initially had amorphous or polycrystalline structures and underwent restructuring through kinetic water and ion mobilization in the presence of sufficient condensed water, resulting in overcoming the kinetic barrier and crystallization during humidification.

Efflorescence during humidification was also observed for type 3 MBTCA particles, in addition to the observation in Fig. 3 for tri-sodium MBTCA particles, which seemed to be attributed to the similar structure to pure MBTCA particles when all three COOH were replaced by COONa. The explanation of "Efflorescence during humidification" will be further explained in the pure MBTCA particles section.

\* Based on the chemical reactions the current study has proposed, acidic conditions are preferable for the reactions to proceed. However, Figure 7b shows the higher

degree of 2 reaction for particles with MBTCA:NaCl = 1:3 than for those with that = 1:1. Particles with the higher abundance of NaCl (MBTCA:NaCl=1:3) are supposed to be less acidic. Do you have any explanation on it?

**Response:** The reactions were driven by the liberation of HCl(g), so the increased availability of both dissociated H+ and Cl- should facilitate the reaction, which makes the degree of reaction follows the sequence of MBTCA:NaCl = 1:3>1:2>1:1. We did not say that acidic conditions are preferable for the reactions to proceed, instead we said about "the availability". The H+ cannot fully dissociate due to the weak acidic property of MBTCA, and NaCl can be fully consumed only when enough H+ is provided, so that the reaction was complete only for the mixtures with MBTCA:NaCl = 2:1 in this study.

\* Figure 8a exhibits some heterogeneity of elemental distributions for particles with 1:2 and 1:3, which makes me wonder where Raman spectra were obtained. Raman spectra obtained from the edge and the center of the particles would be different.

**Response:** The Raman spectra shown in the figures were all obtained in the center of the particles. Actually, the Raman spectra were obtained both at the center and the edge of the particles for comparison during the measurement when the heterogeneity appeared. The reviewer is correct in that the spectra from the edge and the center were different, but only in the intensity since NaCl is Raman inactive. As shown in the figure below (will be put into the supporting information as Fig. S3), the Raman spectra which were obtained from the center and the edge point of an exemplar MBTCA:NaCl = 1:3 particle during the humidification process, corresponds very well after normalization to the CH peak at 1460cm-1. The discussion on the Raman spectra will be given in the revised text.

\* The levitated particles show the lower reactivity than the deposited particles. The authors ascribed it to less release of HCl and the quenching process, which are too speculative. One more factor I want to point out is the almost one order of magnitude difference of the size between the two particles. It may not affect the reaction rates at a given RH, but it may do the release of HCl. I suggest the authors to discuss the potential size effect on the reactivity difference.

**Response:** The default setting of  $N_2$  flow inside the see-through impactor cell and the levitation cell were 4 and 0.2 L·min-1, respectively, so we think that the

levitation system is relatively closed compared to the see-through impactor system, which leads to less release of HCl(g). And the quenching process also dramatically decreases the reaction time due to the crystallization of both organic and inorganic compositions. We agree with the reviewer that the larger size of the levitated particles can also limit the release of HCl (Kerminen et al., 1997). The information will be added into the revised text.

\* Interestingly, when the authors used the levitated particles, no size decrease (i.e., A/A0
- Kerminen, V.-M., Pakkanen, T. A., and Hillamo, R. E.: Interactions between inorganic trace gases and supermicrometer particles at a coastal site, Atmospheric Environment, 31, 2753-2765, 1997.
- Li, X., Gupta, D., Eom, H.-J., Kim, H., and Ro, C.-U.: Deliquescence and efflorescence behavior of individual NaCl and KCl mixture aerosol particles, Atmos. Environ., 82, 36–43, doi:10.1016/j.atmosenv.2013.10.011, 2014.
- Pöhlker, C., Saturno, J., Krüger, M. L., Förster, J.-D., Weigand, M., Wiedemann, K. T., Bechtel, M., Artaxo, P., and Andreae, M. O.: Efflorescence upon humidification? X-ray microspectroscopic in situ observation of changes in aerosol microstructure and phase state upon hydration, Geophys. Res. Lett., 41, 2014GL059409, doi:10.1002/2014gl059409, 2014.
- Schlenker, J. C. and Martin, S. T.: Crystallization Pathways of Sulfate-Nitrate-Ammonium Aerosol Particles, J. Phys. Chem. A, 109, 9980–9985, doi:10.1021/jp052973x, 2005.

Figure S2. Hygroscopic curve of pure NaCl particles. The transition RHs recorded during humidification and dehydration processes are marked with arrows in the hygroscopic curves.

Figure S3. Comparison of Raman spectra of an exemplar particle (MBTCA:NaCl = 1:3) obtained at the center and edge of the particle before and after normalization to the CH peak at 1460 cm-1 during humidification (H) process at RH = 5.9% (end of dehydration) and RH = 73.8% (just before deliquescence).

---

## Referee Comment (RC2) · Anonymous Referee #2 · 6 Aug 2020

In this work, the authors investigated the hygroscopicity of laboratory-generated, micrometer-sized, pure MBTCA and its salts using in-situ Raman microspectrometry (RMS). The authors have clearly demonstrated how interactions between the MBTCA and NaCl could modify the aerosol composition and subsequent hygroscopic behaviors during the hygroscopic measurement. The paper is generally well written. I have minor comments and suggestions for authors' consideration before the paper is accepted for publication.

Comments Line 115, "he particles on the Si wafer were exposed to a hygroscopic measurement cycle, where they experienced dehydration process first (by decreasing

[Figure]

[Figure]

RH from ∼95 to ∼1%), followed by a humidification process (by increasing RH from ∼1 to ∼95%). Can the authors discuss whether an equilibrium state was achieved in their measurements?

Line 139, "Mono-/di-/tri-sodium MBTCA salt solutions were obtained by mixing MBTCA and NaOH (>99.9% purity, Sigma-Aldrich) with molar ratios of MBTCA:NaOH = 1:1, 1:2, and 1:3, respectively." What are the pHs of these aerosols?

Line 251, "The different behavior of the MBTCA particles can be attributed to different nucleation mechanisms, i.e., homogeneous and heterogeneous nucleation, for pure and impure (seed-containing) MBTCA particles, respectively. A similar situation was reported for NH4NO3, NaNO3, and NH4HSO4 particles (Lightstone et al., 2000; Hoffman et al., 2004; Gibson et al., 2006; Kim et al., 2012; Jing et al., 2018; Sun et al., 2018; Wu et al., 2019b). The Si substrates used in this study could also facilitate efflorescence (Eom et al., 2014; Wang et al., 2017)." While the authors have provided possible explanations for the different hygroscopic behaviours of MBTCA particles, it would be important to provide more detailed explanation for each hygroscopic behavior. For example, why did type 1 and type 2 aerosols effloresce at different RH? Any reason why type 3 aerosols were being observed in this work?

Line 284, "NaH2M and Na2HM particles still showed the same shapes and Raman spectra with those at RHs = 3.4% and 2.8%, 285 respectively. These results indicate the non-crystallizable properties and supersaturated amorphous phase state of the particles. The Na3M particles behaved differently as they did not crystallize during the dehydration process. On the other hand, the aerosols exhibited efflorescence at RH = 46.8% during the humidification process (Fig. 3(c)), deliquesced to become a droplet at RH = 53.1%, and grew continuously after that with increasing RH." As mentioned above, can the authors comment:Could the crystallisation of Na3M aerosols attribute to the presence of impurities, the use of Si substrates or other factors? Could the authors rule out these possibilities in their measurements? Can the authors also comment: What is the water uptake (e.g. aerosol water content) of mono-/di-/tri-sodium MBTCA

salt aerosols? Are they different?

Figure 4, unlike the pure MBTCA and mono-/di-/tri-sodium MBTCA salt aerosols, why the dehydration and humidification curves of MBTCA:NaCl = 1:1 and 2:1 do not overlap with each other?

Figure 5, why the dehydration and humidification curves of MBTCA:NaCl = 1:2 and 1:3 do not overlap with each other after deliquescence?

In 3.3.3 Chemical reactivity of aerosols generated from MBTCA−NaCl mixture solutions, the authors have provided a nice discussion of how the chemical composition would evolve due to the chemical reactions between MBTCA and NaCl during the experiments. My question is: do the extent of the reactions depends on the experimental time? Would the duration of the experiments would significantly affect the hygroscopic behaviors observed for different systems in the experiments? Since the chemical compositions evolves over time, can the authors discuss whether an equilibrium state was achieved in their hygroscopic measurements?

Line 472, "Two types of hygroscopic behavior of pure MBTCA particles were observed, corresponding closely to types 1 and 3 aerosol particles in the see-through impactor system." Can the authors further elaborate why type 1 aerosol is observed in levitated pure MBTCA aerosols?

―――――――――――――――――――――

---

## Author Response (AR1)

**General Comment from Anonymous Referee #1**

*Li et al. report for the first time the hygroscopic measurements of MBTCA-inorganic system using Raman spectroscopy. The reactions involving HCl(g) release were discussed using the speciation of COOH and COO⁻, and related to the hygroscopic curves obtained by the optical imaging of the nominal particle size. This work would be an important increment to our knowledge of SOA hygroscopicity, but the current version is not ready for publication in ACP because of a number of issues as shown* below.

**Response:** We thank the reviewer very much for the careful evaluation and valuable comments for our work. We revised the manuscript as much as possible respecting the reviewer's comments.

**Specific comments from Anonymous Referee #1 (comments are in italic)**

*\* The abstract is rather long. I suggest the authors to make the text more concise by focusing more results and their interpretation.*

**Response:** We revised the abstract in a more concise way as can be seen in the revised abstract.

*\* Lines 82-83: It is not clear how MBTCA can accelerate the new particle formation. The authors mentioned that MBTCA is extremely low volatile and one of the SOA components, i.e., one of the particulate compositions. Are you saying that preexisting aerosols enhance the new particle formation?*

**Response:** As described in the papers (Donahue et al., 2013; Elm, 2019), MBTCA can nucleate the sulfuric acid vapor or provide polar functional groups to form and grow molecular clusters through hydrogen bonds, which promote the new particle formation. However, as this sentence is not directly related to our work, it was omitted in the revised version without deteriorating the context.

*\* Line 127: I don't agree with the author's statement that RMS can provide the chemical compositions. RMS detects chemical functional groups of a molecule of interest.*

**Response:** The reviewer is correct, and the sentence was modified as "RMS can provide information on chemical functional groups, water contents, molecular interactions, and phase states of the aerosol particles" (lines 133-135 in the marked revised version below).

*\* Lines 170-173: It is useful if the authors show the experimentally measured hygroscopic curves of NaCl together with the thermodynamic model prediction.*

**Response:** The hygroscopic curve of NaCl (as shown below) was put into the supporting information as Fig S2.

*\* Why were deposited and levitated particles examined in this study. The authors need to elaborate the necessity of using levitated particles.*

**Response:** "The deposited particles (~6.5 µm in average in this study) may have some influences from the collecting substrate such as a facilitated heterogeneous nucleation, which can be eliminated in the levitation system due to the substrate-free and contactless properties. However, the particles in the levitation system are generally large in size (~80 µm in average in this study), which is less atmospherically relevant. And thus the analysis of the particles in both systems is expected to give more detailed information on the hygroscopic behavior of MBTCA aerosols." This paragraph was added in the revised text (lines 119-125).

*In Figure 2, the authors need to explicitly define the type of MBTCA particles. How do types 1-3 of the particles differ?*

  **Response:** The following argument was added in the marked revised manuscript (lines 271-295). "Specifically, the types of pure MBTCA particles were classified as, "type 1: with a prompt efflorescence at ~50% RH during dehydration; type 2: with a gradual efflorescence at ~35% RH during dehydration; type 3: with a gradual efflorescence at ~37% RH during humidification", based on their different behavior when efflorescence occurred. The different efflorescence behavior was attributed to different nucleation mechanisms: heterogeneous nucleation for types 1 and 2 particles (seed-containing), and homogeneous nucleation for type 3 particles (pure). MBTCA powders, which was used for making the MBTCA solution, has intrinsic unknown impurity of 2%, and they were used without any purification. When MBTCA powders were dissolved in DI water and particles were generated by the nebulization of the aqueous solution using $N_2$ gas, impurities were either absent or associated with the droplets. The impurities existed in types 1 and 2 particles after nebulization, acting as seed crystals to induce efflorescence. Aqueous moieties in particles were reported to effloresce more easily by heterogeneous nucleation in the presence of seeds (Schlenker and Martin, 2005; Li et al., 2014; Gupta et al., 2015). The lower ERH and gradual efflorescence of type 2 compared to type 1 particles might be due to the less amount of impurities. Type 3 particles contain negligible or no seed crystals, and large kinetic barrier and/or diffusional resistance make the formation of the crystal structure difficult owing to the decreasing availability of condensed water during dehydration, so that they did not experience any efflorescence. A similar situation was observed for NH4NO3, NaNO3, and NH4HSO4 particles (Lightstone et al., 2000; Hoffman et al., 2004; Gibson et al., 2006; Kim et al., 2012; Jing et al., 2018; Sun et al., 2018; Wu et al., 2019b). The Si substrates used in this study could also facilitate efflorescence (Eom et al., 2014; Wang et al., 2017). The efflorescence during humidification like type 3 particles was previously reported for Amazonian rain forest aerosols (Pöhlker et al., 2014) and the laboratory-generated NaCl–MgCl2 mixture particles (Gupta et al., 2015). Thus, this phenomenon is not rare. And it was claimed that the aerosol particles initially had amorphous or poly-crystalline structures and underwent restructuring through kinetic water and ion mobilization in the presence of sufficient condensed water, resulting in overcoming the kinetic barrier and crystallization during humidification."

*Line 230: Please provide the evidence of the phase change at RH = 36.7%. The optical images of type 3 particle in Figure 2 show the morphology change from 34% RH to 36.7% RH, but it does not necessarily mean phase change.*

**Response:** As shown in the optical image and Raman spectrum in Fig. 2, type 3 particles became irregular in shape during the humidification process, and the overlapped C=O (from COOH) peak at 1660 - 1680 $cm^{-1}$ entirely shifted to 1660 $cm^{-1}$ when the RH was increased from 34% to 36.7%. Both the morphological and RMS data clearly indicate the phase change from amorphous/solid-state to solid state. The sentence at line 230 was modified as, "type 3 particles experienced a gradual morphological change at RH = 34-36.7% and remained the same until RH = ~85%", for the precise expression (line 240 in the marked revised version).

*\* Lines 250-252: Please clarify if the authors purposely injected impurity in the MBTCA particles. This is related to my earlier question: what is the type?*

**Response:** The MBTCA powders with intrinsic 2% impurity were used without any purification and no impurity was injected purposely. The information was clarified in the revised text (lines 276-277 in the marked revised version). Regarding the types of the particles, please see the response above.

*\* Line 270: It is confusing. The implication is based on the previous study or the current study?*

**Response:** The sentence was modified as, "A previous study showed that MBTCA was not hydrated significantly in the ambient atmosphere (Kildgaard et al., 2018), and our results also implied that the MBTCA stay in the air as solids once they effloresced." (line 300 in the marked revised version), to avoid the confusion.

*\* In Figure 3, one of the interesting results is the efflorescence at 46.8% RH during the humidification. The explanation for this phenomenon is rather dry. The authors need to elaborate more. It would be also useful if the authors can discuss the repeatability of such a phenomenon.*

**Response:** Efflorescence during humidification was previously reported for Amazonian rain forest aerosols (Pöhlker et al., 2014) and the laboratory-generated NaCl–$MgCl_2$ mixture particles (Gupta et al., 2015). And thus this phenomenon is not rare and it was claimed that the aerosol particles initially had amorphous or poly-crystalline structures and underwent restructuring through kinetic water and ion mobilization in the presence of sufficient condensed water, resulting in overcoming the kinetic barrier and crystallization during humidification. ← This paragraph was added (lines 290-295 in the marked revised version)

Efflorescence during humidification was also observed for type 3 MBTCA particles, in addition to the observation in Fig. 3 for tri-sodium MBTCA particles, which seemed to be attributed to the similar structure to pure MBTCA particles when all three COOH were replaced by COONa.

*Based on the chemical reactions the current study has proposed, acidic conditions are preferable for the reactions to proceed. However, Figure 7b shows the higher degree of 2 reaction for particles with MBTCA:NaCl = 1:3 than for those with that = 1:1. Particles with the higher abundance of NaCl (MBTCA:NaCl=1:3) are supposed to be less acidic. Do you have any explanation on it?*

**Response:** The reactions were driven by the liberation of HCl(g), so the increased availability of both dissociated $H^+$ and $Cl^-$ should facilitate the reaction, which makes the degree of reaction follows the sequence of MBTCA:NaCl = 1:3>1:2>1:1. We did not say that acidic conditions are preferable for the reactions to proceed, instead we said about "the availability". The $H^+$ cannot fully dissociate due to the weak acidic property of MBTCA, and NaCl can be fully consumed only when enough $H^+$ is provided, so that the reaction was complete only for the mixtures with MBTCA:NaCl = 2:1 in this study.

*Figure 8a exhibits some heterogeneity of elemental distributions for particles with 1:2 and 1:3, which makes me wonder where Raman spectra were obtained. Raman spectra obtained from the edge and the center of the particles would be different.*

**Response:** The Raman spectra shown in the figures were all obtained in the center of the particles. However, the Raman spectra were obtained both at the center and the edge of the particles for comparison during the measurement when the heterogeneity appeared. The reviewer is correct in that the spectra from the edge and the center were different, but only in the intensity since NaCl is Raman inactive. As shown in the figure below (which was put into the supporting information as Fig. S3), the Raman spectra which were obtained from the center and edge of an exemplar MBTCA:NaCl = 1:3 particle during the humidification process, match well after normalization to the CH peak at $1460cm^{-1}$. The discussion on the Raman spectra was given (lines 398-404 in the marked revised version).

*\* The levitated particles show the lower reactivity than the deposited particles. The authors ascribed it to less release of HCl and the quenching process, which are too speculative. One more factor I want to point out is the almost one order of magnitude difference of the size between the two particles. It may not affect the reaction rates at a given RH, but it may do the release of HCl. I suggest the authors to discuss the potential size effect on the reactivity difference.*

**Response:** The default setting of $N_2$ flow inside the see-through impactor cell and the levitation cell were 4 and 0.2 L·min$^{-1}$, respectively, so we think that the levitation system is relatively closed compared to the see-through impactor system, which leads to less release of HCl(g). And the quenching process also dramatically decreases the reaction time due to the crystallization of both organic and inorganic compositions. We agree with the reviewer that the larger size of the levitated particles can also limit the release of HCl (Kerminen et al., 1997). The information was added into the marked revised text (lines 553-557 in the marked revised version).

*\* Interestingly, when the authors used the levitated particles, no size decrease (i.e., A/A0 < 1) was observed. The authors need to explain it. The size decrease observed with the deposited particles was attributed to the structural rearrange in the main text, but we may be seeing the effect of substrate.*

**Response:** We think why the shrinkage of the particles before deliquescence was not captured for the levitated particles can be because the size of the particles (~80 μm in average in this study) is too large for the structural rearrangement to be observed. And the 2-D optical images of the particles were used for plotting the hygroscopic curves even though the particles were levitated, which might lead to some missing information on the 3-D level. ← (lines 534-538 in the marked revised version) We agree with the reviewer that the substrate can also affect the shrinkage. Especially, the hydrophilic substrate (such as Si wafer used in this study) seems to favor this phenomenon (Eom et al., 2014).

Figure S2. Hygroscopic curve of a pure NaCl particle. The transition RHs recorded during humidification and dehydration processes are marked with arrows in the hygroscopic curves.

[Figure]

Figure S3. Comparison of Raman spectra of an exemplar particle (MBTCA:NaCl = 1:3) obtained at the center and edge of the particle before and after normalization to the CH peak at 1460 cm$^{-1}$ during humidification (H) process from RH = 5.9% (end of dehydration) to RH = 73.8% (just before deliquescence).

[Figure]

**General Comment from Anonymous Referee #2**

*In this work, the authors investigated the hygroscopicity of laboratory-generated, micrometer-sized, pure MBTCA and its salts using in-situ Raman microspectrometry (RMS). The authors have clearly demonstrated how interactions between the MBTCA and NaCl could modify the aerosol composition and subsequent hygroscopic behaviors during the hygroscopic measurement. The paper is generally well written. I have minor comments and suggestions for authors' consideration before the paper is accepted for publication.*

    **Response:** We thank the reviewer very much for the careful evaluation and valuable comments for our work. Here we provide the response to the reviewer's comments.

**Specific comments from Anonymous Referee #2 (comments are in italic)**

*\* Line 115, "The particles on the Si wafer were exposed to a hygroscopic measurement cycle, where they experienced dehydration process first (by decreasing RH from ~95 to ~1%), followed by a humidification process (by increasing RH from ~1 to ~95%). Can the authors discuss whether an equilibrium state was achieved in their measurements?*

    **Response:** During the measurements, each humidity condition (from ~1 to ~95% RH with 1% RH step) was sustained for at least 2 mins in order to provide an enough time for condensing or evaporating of water. As the reaction of MBTCA and NaCl is driven by the gaseous HCl liberation in the open hygroscopic measurement system, the equilibrium cannot be achieved. The information was added in the revised version (lines 174-176 in the marked revised version).

*\* Line 139, "Mono-/di-/tri-sodium MBTCA salt solutions were obtained by mixing MBTCA and NaOH (>99.9% purity, Sigma-Aldrich) with molar ratios of MBTCA:NaOH = 1:1, 1:2, and 1:3, respectively." What are the pHs of these aerosols?*

**Response:** The initial pHs of the nebulized aerosol particles should be equal to the solution (0.3 M for all the solutions). The roughly calculated pH values without considering temperature and activity of the chemicals are 2.04, 4.22, 5.82, and 10.16, while the measured pH values are 2.03, 4.07, 5.66, and 10.73 for MBTCA, mono-/di-/tri-sodium MBTCA salt solutions, respectively. The pHs of the aerosols according to RH changes are not available in our study.

*\* Line 251, "The different behavior of the MBTCA particles can be attributed to different nucleation mechanisms, i.e., homogeneous and heterogeneous nucleation, for pure and impure (seed-containing) MBTCA particles, respectively. A similar situation was reported for $NH_4NO_3$, $NaNO_3$, and $NH_4HSO_4$ particles (Lightstone et al., 2000; Hoffman et al., 2004; Gibson et al., 2006; Kim et al., 2012; Jing et al., 2018; Sun et al., 2018; Wu et al., 2019b). The Si substrates used in this study could also facilitate efflorescence (Eom et al., 2014; Wang et al., 2017)." While the authors have provided possible explanations for the different hygroscopic behaviors of MBTCA particles, it would be important to provide more detailed explanation for each hygroscopic behavior. For example, why did type 1 and type 2 aerosols effloresce at different RH? Any reason why type 3 aerosols were being observed in this work?*

**Response:** Please see the response to the Referee #1 regarding the further explanations of particle types.

*\* Line 284, "NaH$_2$M and Na$_2$HM particles still showed the same shapes and Raman spectra with those at RHs = 3.4% and 2.8%, respectively. These results indicate the non-crystallizable properties and supersaturated amorphous phase state of the particles. The Na$_3$M particles behaved differently as they did not crystallize during the dehydration process. On the other hand, the aerosols exhibited efflorescence at RH = 46.8% during the humidification process (Fig. 3(c)), deliquesced to become a droplet at RH = 53.1%, and grew continuously after that with increasing RH." As mentioned above, can the authors comment: Could the crystallization of Na$_3$M aerosols attribute to the presence of impurities, the use of Si substrates or other factors? Could the authors rule out these possibilities in their measurements? Can the authors also comment: What is the water uptake (e.g. aerosol water content) of mono-/di-/tri-sodium MBTCA salt aerosols? Are they different?*

**Response:** As explained in the paper, the 3 types of pure MBTCA particles with different behavior were observed in the same hygroscopic measurement among 100 particles on the substrate, and crystallization of types 1 and 2 were attributed to the random inclusion of impurity seed particles or the Si wafer substrate,  while that of type 3 was due to homogeneous crystallization. However, as for the Na$_3$M aerosols, all the observed particles on the substrate (around 100 particles) behaved exactly in the same way as shown in the Fig. 3(c) with crystallization during humidification process, similar to the type 3 pure MBTCA particles, so we think the major contribution to the crystallization of Na$_3$M particles is also homogeneous crystallization, instead of heterogeneous crystallization induced by impurities and Si substrate effect. Please see the response to the Referee #1's comment which is similar to this comment.

The quantification of the aerosol water content is not available in our study. But based on the water band in the Raman spectra at ~3400 cm$^{-1}$ in Fig. 3, the water content is in the sequence of tri- > di- > mono-sodium MBTCA droplets.

*\* Figure 4, unlike the pure MBTCA and mono-/di-/tri-sodium MBTCA salt aerosols, why the dehydration and humidification curves of MBTCA:NaCl = 1:1 and 2:1 do not overlap with each other?*
*\* Figure 5, why the dehydration and humidification curves of MBTCA:NaCl = 1:2 and 1:3 do not overlap with each other after deliquescence?*

**Response:** These two comments will be responded together. The humidification and dehydration curves of the MBTCA-NaCl mixtures do not overlap with each other mainly due to the different amounts of NaCl in the processes. As shown in the Fig. S2 above, NaCl is quite hygroscopic with around four times change in 2-D area after deliquescence, so the decreased amount of NaCl in the mixtures also lead to the smaller 2-D area when the particles of MBTCA:NaCl = 1:1 and 2:1 and MBTCA:NaCl = 1:2 and 1:3 experienced hygroscopic growth during the humidification process, compared to those before dehydration. The information was added into the text (lines 419-424 in the marked revised version).

*\* In 3.3.3 Chemical reactivity of aerosols generated from MBTCA−NaCl mixture solutions, the authors have provided a nice discussion of how the chemical composition would evolve due to the chemical reactions between MBTCA and NaCl during the experiments. My question is: do the extent of the reactions depends on the experimental time? Would the duration of the experiments would significantly affect the hygroscopic behaviors observed*

*for different systems in the experiments? Since the chemical compositions evolves over time, can the authors discuss whether an equilibrium state was achieved in their hygroscopic measurements?*

**Response:** As responded above, each RH point was sustained for at least 2 mins, but the equilibrium cannot be archived as the measurement system is not closed. It is natural to assume that the experimental time has effects on the reaction extent under the open measurement system. As discussed for the levitation experiment which uses a somewhat more closed system than the see-through impactor system, the reaction extents were smaller than those for the see-through impactor system.

*\* Line 472, "Two types of hygroscopic behavior of pure MBTCA particles were observed, corresponding closely to types 1 and 3 aerosol particles in the see-through impactor system." Can the authors further elaborate why type 1 aerosol is observed in levitated pure MBTCA aerosols?*

**Response:** As described in the paper, the levitated droplets experienced quenching process before the hygroscopic cycle measurement. During the quenching and humidification processes, two types of hygroscopic behavior of pure MBTCA particles were observed, corresponding closely to types 1 and 3 aerosol particles in the see-through impactor system, due to heterogeneous crystallization induced by impurity seed crystals and homogeneous crystallization, respectively. Since the levitation system is substrate-free, the substrate effect is eliminated. The information was added into the text (lines 523-524 in the marked revised version).

[revised manuscript text omitted]

---

## Author Response (AR2)

**General Comment from Anonymous Referee #1**

Thank the authors for putting the efforts in the revision of the text. The authors adequately responded most questions, but I have still some comments and suggestions. The text would be ready for publication in ACP after the following is addressed.

**Response:** We thank the reviewer very much for the careful evaluation and valuable comments for our work. We revised the manuscript as much as possible respecting the reviewer's comments.

**Specific comments from Anonymous Referee #1 (comments are in italic)**

*\* I believe that the abstract is still long. There are still the experimental information remaining.*

**Response:** We revised the abstract more concisely as can be seen in the revised abstract.

*\* About the type of particles (lines 271–295), I think that the authors are talking about "the type of hygroscopicity" during the cycle between humidification and dehumidification. Three particles were generated from the same stock solution, but the authors observed three different behaviors in the hygroscopicity. Using the wording of particle type may mislead us to think of three types of particles prepared by three different sources.*

**Response:** We agree with the reviewer that we are talking about "the type of hygroscopicity" of particles during the hygroscopic process, and as seen in the beginning of the section 3.1, we have explained that "Wet-deposited MBTCA aerosols exhibited three types of hygroscopic behaviors". In order not to cause any misleading, the sentence will be modified as below: "Wet-deposited MBTCA aerosols generated by the nebulization from a pure MBTCA solution exhibited three different types of hygroscopic behaviors, which are termed as "types 1, 2, and 3". (lines 225-226 in the marked revised version below).

The sentence at line 264 will be modified as "Specifically, the types of hygroscopicity of pure MBTCA particles were classified as".

*It seems that the authors overlooked my question on" the repeatability of the phenomenon of efflorescence during the humidification". Can you respond this question?*

**Response:** As explained at lines 317-321 in the marked revised version below, all the observed $Na_3M$ particles on the substrate (around 100 particles) behaved exactly in the same way with crystallization during humidification process and we think that this phenomenon is repeatable. And we've also explained at lines 283-285 in the marked revised version below that the efflorescence during humidification is not rare as it was also observed by other studies.

*I am not sure if the large size of the particles used in the levitation is the reason why the particle shrinkage could not be captured. The size change captured by a camera should be more apparent for large particles because the particle shrinkage is defined as the relative change in particle size. Let's say a few percent size decrease for the observation of the shrinkage relative to the particle size at low RH. The absolute size change should be larger for larger particles so that the size change can be more visible.*

**Response:** We agree with the reviewer. Actually, the optical image quality of the levitated particles was not as good as those in the see-through impactor system, which made the 2-D area plot less accurate and precise. However, the shrinkage of the particles before the deliquescence is not much important in our study and does not influence our observations regarding the hygroscopicity of the pure and mixture particles. We deleted the sentences at line 527-531 in the marked revised version below without deteriorating the context.

*Minor point:*

*Line 28: dissolve -> deliquesce*

**Response:** Modified as suggested (line 24 in the marked revised version).

[revised manuscript text omitted]